# Hemozoin produced by mammals confers heme tolerance

Rini H Pek[1,2], Xiaojing Yuan[1,2], Nicole Rietzschel[1,2], Jianbing Zhang[1,2], Laurie Jackson[3], Eiji Nishibori[4,5], Ana Ribeiro[1,2], William Simmons[6], Jaya Jagadeesh[6], Hiroshi Sugimoto[7], Md Zahidul Alam[8], Lisa Garrett[9], Malay Haldar[8], Martina Ralle[10], John D Phillips[3], David M Bodine[6], Iqbal Hamza[1,2]*

[1]Department of Animal and Avian Sciences, University of Maryland, College Park, United States; [2]Department of Cell Biology and Molecular Genetics, University of Maryland, College Park, United States; [3]Department of Medicine, University of Utah School of Medicine, Salt Lake City, United States; [4]Faculty of Pure and Applied Sciences, University of Tsukuba, Tsukuba, Japan; [5]Tsukuba Research Center for Energy Materials Science, University of Tsukaba, Tsukaba, Japan; [6]Genetics and Molecular Biology Branch, National Human Genome Research Institute, National Institutes of Health, Bethesda, United States; [7]RIKEN SPring-8 Center, Sayo, Hyogo, Japan; [8]Department of Pathology and Laboratory Medicine, Perelman School of Medicine at the University of Pennsylvania, Philadelphia, United States; [9]NHGRI Embryonic Stem Cell and Transgenic Mouse Core, National Human Genome Research Institute, National Institutes of Health, Bethesda, United States; [10]Department of Molecular and Medical Genetics, Oregon Health and Science University, Portland, United States

*For correspondence:
hamza@umd.edu

**Abstract** Free heme is cytotoxic as exemplified by hemolytic diseases and genetic deficiencies in heme recycling and detoxifying pathways. Thus, intracellular accumulation of heme has not been observed in mammalian cells to date. Here we show that mice deficient for the heme transporter SLC48A1 (also known as HRG1) accumulate over ten-fold excess heme in reticuloendothelial macrophage lysosomes that are 10 to 100 times larger than normal. Macrophages tolerate these high concentrations of heme by crystallizing them into hemozoin, which heretofore has only been found in blood-feeding organisms. *SLC48A1* deficiency results in impaired erythroid maturation and an inability to systemically respond to iron deficiency. Complete heme tolerance requires a fully-operational heme degradation pathway as haplo insufficiency of *HMOX1* combined with *SLC48A1* inactivation causes perinatal lethality demonstrating synthetic lethal interactions between heme transport and degradation. Our studies establish the formation of hemozoin by mammals as a previously unsuspected heme tolerance pathway.

## Introduction

Hemolysis can arise from genetic mutations (*Larsen et al., 2012*), parasitic and bacterial infections (*Orf and Cunnington, 2015*), and drug-induced autoimmune reactions (*Garratty, 2009*). Rupturing of red blood cells (RBCs) releases high concentrations of heme, which promotes reactive oxygen species (ROS) production (*Fibach and Rachmilewitz, 2008*) and inflammation (*Dutra et al., 2014*) causing tissue damage and death. Consistent with the cytotoxicity of heme, genetic ablation of the heme degrading enzyme, heme oxygenase 1 (HMOX1), results in >90% embryonic lethality (*Kovtunovych et al., 2010*; *Poss and Tonegawa, 1997a*; *Poss and Tonegawa, 1997b*). The few

**eLife digest** Specialized cells, known as red blood cells, are responsible for transporting oxygen to various organs in the body. Each red blood cell contains over a billion molecules of heme which make up the iron containing portion of the hemoglobin protein that binds and transports oxygen. When red blood cells reach the end of their life, they are degraded, and the heme and iron inside them is recycled to produce new red blood cells. Heme, however, is highly toxic to cells, and can cause severe tissue damage if not properly removed.

Scavenger cells called macrophages perform this recycling role in the spleen, liver and bone marrow. Collectively, macrophages can process around five million red blood cells every second or about 100 trillion heme molecules. But, it is unclear how they are able to handle such enormous volumes.

Macrophages isolated from human and mice have been shown to transport heme from damaged red blood cells using a protein called HRG1. To investigate the role HRG1 plays in heme-iron recycling, Pek et al. used a gene editing tool known an CRISPR/Cas9 to remove the gene for HRG1 from the macrophages of mice. If HRG1 is a major part of this process, removing the gene should result in a build-up of toxic heme and eventual death of the mouse.

But, rather than dying of heme-iron overload as expected, these mutant mice managed to survive. Pek et al. found that despite being unable to recycle heme, these mice were still able to make new red blood cells as long as they had a diet that was rich in iron. However, the darkening color of the spleen, bone marrow, and liver in these HRG1 deficient mice indicated that these mice were still accumulating high levels of heme. Further experiments revealed that these mice protected themselves from toxicity by converting the excess heme into crystals called hemozoin. This method of detoxification is commonly seen in blood-feeding parasites, and this is the first time it has been observed in a mammal.

These crystals invite new questions about how mammals recycle heme and what happens when this process goes wrong. The next step is to ask whether humans also start to make hemozoin if the gene for HRG1 is faulty. If so, this could open a new avenue of exploration into treatments for red blood cell diseases like anemia and iron overload.

mice that survive are susceptible to hemolytic infections (*Seixas et al., 2009*), and have little or no reticuloendothelial macrophages (*Kovtunovych et al., 2010*), a phenotype attributed to heme cytotoxicity.

The majority of body heme is encapsulated within RBCs, and each RBC contains over one billion heme molecules. As RBCs undergo senescence or become damaged, they are engulfed by macrophages from the reticuloendothelial system (RES) through a process termed erythrophagocytosis (EP) (*Ganz, 2012*; *Ganz and Nemeth, 2006*; *Klei et al., 2017*; *Winter et al., 2014*). As RBCs contain both heme and non-heme iron, transporters for each metabolite are recruited to the erythrophagosomal membranes (*Delaby et al., 2012*; *Soe-Lin et al., 2010*). The heme responsive gene-1 (HRG1/SLC48A1) transports heme from the erythrophagosome into the cytosol (*Rajagopal et al., 2008*; *White et al., 2013*). Heme is enzymatically degraded by HMOX1 (*Kovtunovych et al., 2010*; *Kovtunovych et al., 2014*) to liberate iron, which can either be stored in ferritin (FTH1) or exported out of the cell by ferroportin (SLC40A1) to be reutilized for new RBC production (*Knutson et al., 2005*; *Knutson et al., 2003*), including developing erythroblasts in the bone marrow (*Raja et al., 1999*). Excess heme is exported from the cell by feline leukemia virus subgroup C cellular receptor 1 (FLVCR1) to prevent heme toxicity (*Keel et al., 2008*). Consequently, genetic disruption in *HMOX1*, *SLC40A1*, *FTH1*, and *FLVCR1* - steps in the heme-iron recycling pathway - causes embryonic lethality in mice. Here, we show that mice lacking the heme transporter *SLC48A1* are viable despite accumulating high concentrations of heme. These animals are heme tolerant because they sequester heme within enlarged lysosomes in the RES macrophages and form crystalline hemozoin, which heretofore has only been found in blood-feeding organisms (*Shio et al., 2010*; *Toh et al., 2010*). Our work suggests the existence of a previously unknown pathway for heme detoxification and tolerance in mammals.

## Results

### Reticuloendothelial tissues accumulate dark pigments in the absence of *SLC48A1*

To uncover the in vivo function of SLC48A1 in mammals, we generated *SLC48A1*-deficient mice using CRISPR/Cas9 gene editing. Guide RNAs targeting exon 1 of mouse *SLC48A1* (*Figure 1A*) produced seven mutant alleles in C57BL/6J × 129/SvJ $F_1$ animals (Table 1 in *Supplementary file 1*) which were backcrossed to C57BL/6J mice before intercrossing. We observed similar phenotypes in all mutant alleles and focused on the M6 allele which contains a two base-pair deletion in exon 1 of *SLC48A1* (M6). This deletion causes a frameshift within the thirty-third codon immediately after the first transmembrane domain (*Figure 1B*; *Figure 1—figure supplement 1A*). Intercrossing SLC48A1 HET animals produced KO (knockout) animals with the expected Mendelian ratio (*Figure 1—figure supplement 1B*). While *SLC48A1* mRNA was still detected (not shown), immunoblots and immuno-histochemistry of KO RES tissues showed no detectable SLC48A1 protein, compared to WT (wild-type) tissues which express abundant SLC48A1 (*Figure 1C–D*; *Figure 1—figure supplement 1C–D*). KO mice had significantly larger spleens and lower hematocrits (*Figure 1E,F*). Gross morphological examination of six-week old KO mice revealed darkened spleen, bone marrow, and liver (*Figure 1G*) that corresponded with dark intracellular pigments in histochemical tissue sections (*Figure 1D*, right panel).

### KO mice exhibit extramedullary erythropoiesis with fewer mature RPMs

An enlarged spleen is a common hallmark of ineffective and stress erythropoiesis (*Lenox et al., 2005*; *Perry et al., 2009*). We therefore investigated the erythroid compartment of bone marrows and spleens of these mice using Ter-119 and CD44 markers to follow erythroid differentiation and proliferation (*Chen et al., 2009*). Consistent with the reduced hematocrits, KO mice had fewer total Ter-119$^+$ cells in the bone marrow but more basophilic erythroblasts (population II) which are symptomatic of an impairment in erythroid maturation (*Figure 2A–C*; *Figure 2—figure supplement 1A*). In the spleen, despite similar numbers of total Ter-119$^+$ cells (*Figure 2D,E*), KO animals showed significantly more immature erythroid precursor cells (population II+III), suggesting compensatory erythropoiesis in the spleen (*Figure 2D–F*; *Figure 2—figure supplement 1B*).

Since *SLC48A1* is primarily expressed in RES macrophages (*White et al., 2013*), we analyzed red pulp macrophages (RPMs), which are the primary iron-recycling macrophages in the spleen (*Beaumont and Delaby, 2009*; *Ganz and Nemeth, 2012*; *Haldar et al., 2014*). Significantly fewer mature RPMs (F4/80$^{hi}$Treml4$^+$) were detected in KO spleens (*Figure 2G,H*; *Figure 2—figure supplement 1C*) which correlated with increased numbers of immature RPMs by ratiometric quantification of monocytes (F4/80$^{int}$: F4/80$^{lo}$-CD11b$^{hi}$) (*Figure 2I,J*; *Figure 2—figure supplement 1D*).

### Heme accumulates within RES macrophages of KO mice

KO mice on a standard diet (380 ppm Fe) have normal serum iron, total iron-binding capacity (TIBC) and transferrin saturation but significantly elevated serum ferritin, an indicator of tissue iron-overload (*Ganz and Nemeth, 2012*) (Table 2 in *Supplementary file 1*). Histological analysis by H and E staining showed dark pigmented inclusions accumulating within RES organs of KO mice (*Figure 3A*, right panel). However, in situ Perls' Prussian blue staining did not show significant differences in iron deposition in tissue biopsies (*Figure 3B*). Because it is possible that the dark pigments masked the visualization of the Prussian blue iron complex, we performed inductively coupled plasma mass spectrometry (ICP-MS) to measure total metal content. Significantly more iron was found in the spleens, livers and bone marrows of KO mice (*Figure 3C*), as compared to copper, zinc, and manganese (*Figure 3—figure supplement 1A–D*); a modest 1.4-fold change in manganese was observed in the spleens of KO mice in contrast to 2-fold increase in iron (*Figure 3—figure supplement 1D*). Differences in total metal content were also detected in other organs analyzed (*Figure 3—figure supplement 1A–D*). These results show that KO mice accumulate tissue iron that is not detectable by Prussian blue staining. Through a combination of ultra-performance liquid chromatography (UPLC) (*Sinclair et al., 2001*) and ICP-MS to measure heme-iron and total iron, we observed that KO spleens and bone marrows had significantly more total heme (10-fold and 3-fold, respectively) than

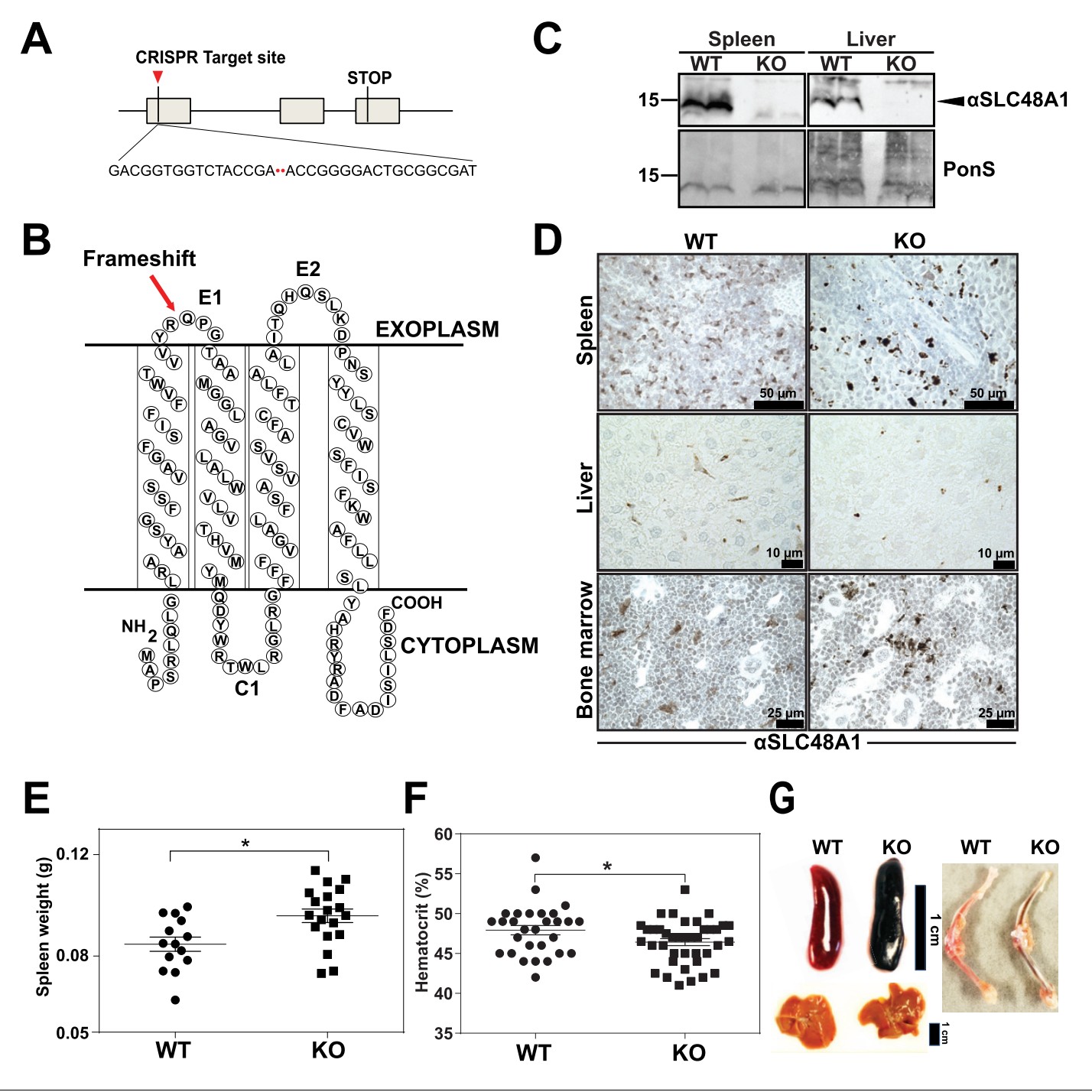

**Figure 1.** Reticuloendothelial tissues accumulate dark pigments in the absence of *SLC48A1*. (A) Structure of the *SLC48A1* gene (which encodes SLC48A1) indicating the CRISPR target site in exon 1. (B) Predicted topology of SLC48A1 protein; arrow indicates the site of the two basepair deletion resulting in frameshift mutation. (C) Immunoblot analysis of membrane lysates prepared from spleens and livers of mice. Membranes were probed with anti-SLC48A1 antibody and then incubated with HRP-conjugated anti-rabbit secondary antibody. Each lane represents one animal. (D) SLC48A1 immunohistochemistry analysis of paraffin-embedded tissue sections of mice. Tissue sections were probed with affinity-purified anti-SLC48A1 antibody and then incubated with HRP-conjugated anti-rabbit secondary antibody. Images shown are representative of at least three mice. (E–F) Spleen wet weights and whole blood hematocrit from WT and KO mice. Each dot represents one mouse; mice were age (6 weeks) and sex-matched. (G) Representative images of spleens, livers and bone marrows of age and sex-matched mice. *p<0.05.
The online version of this article includes the following figure supplement(s) for figure 1:

*Figure 1 continued on next page*

*Figure 1 continued*

**Figure supplement 1.** Genetic lesion in *SLC48A1*, genotypic segregation and loss of SLC48A1 IHC by alkaline-phosphatase.

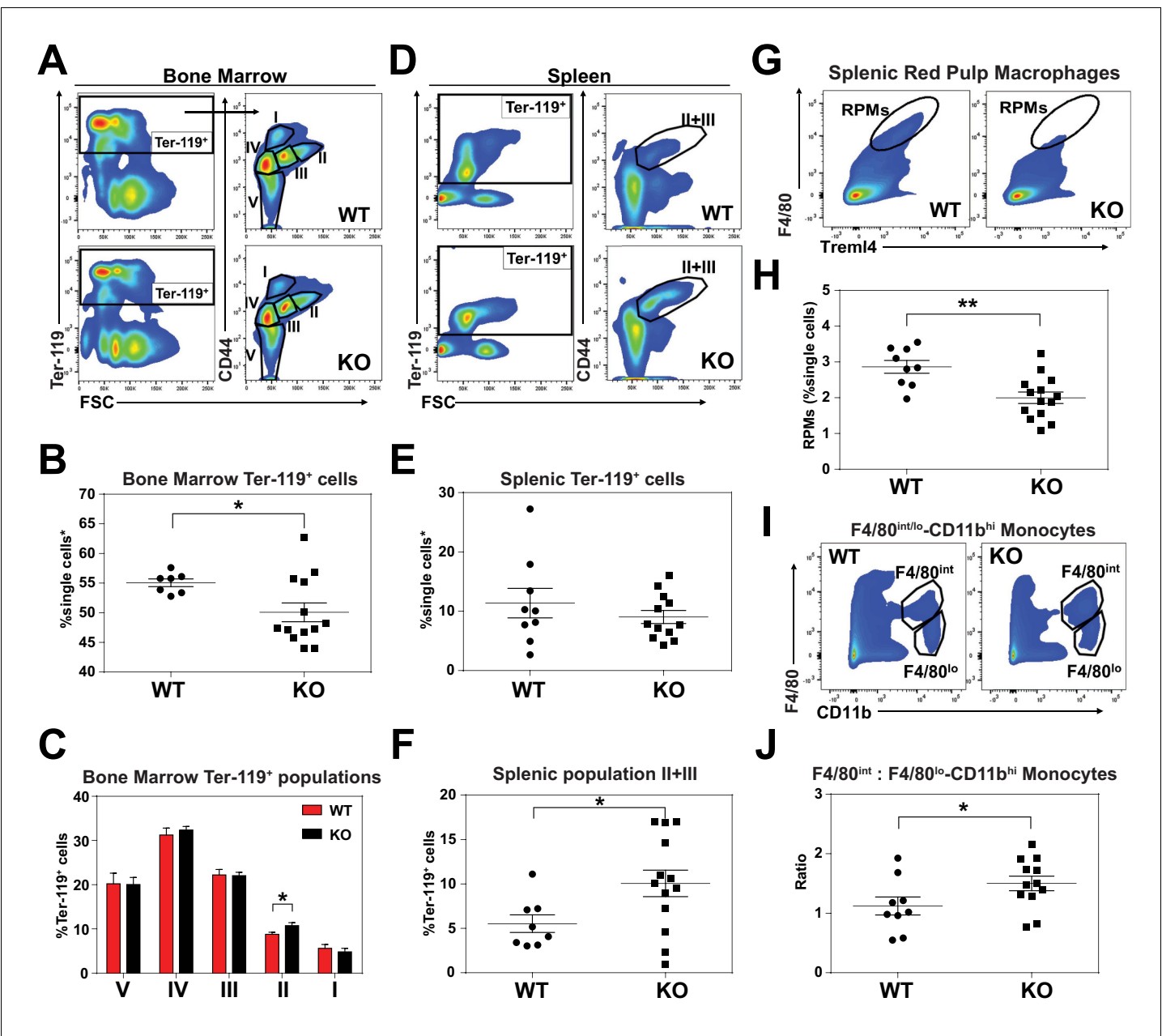

**Figure 2.** KO mice exhibit extramedullary erythropoiesis with fewer mature RPMs Gating strategy of Ter-119+ cells in the (**A**) bone marrow and (**D**) spleen. Quantifications of total Ter-119+ cells in the (**B**) bone marrow and (**E**) spleen. The %single cells* on the y-axis denote single cells that are negative for CD4/8/41, B220 and Gr-1. (**C**) Quantification of subpopulations of Ter-119+ cells represented as a percentage of total Ter-119+ cells in the bone marrow (n = 7–12). (**F**) Quantification of populations II and III of Ter-119+ cells represented as a percentage of total Ter-119+ cells in the spleen. Gating strategy (**G, I**) and quantification (**H, J**) of (**G, H**) splenic F4/80hiTreml4+red pulp macrophages (RPMs) and (**I, J**) F4/80hi and F4/80lo-CD11bhi splenic monocytes. At least 100,000 single cells were analyzed per sample. Each dot represents one mouse. **p<0.05; **p<0.01.
The online version of this article includes the following figure supplement(s) for figure 2:

**Figure supplement 1.** Representative flow cytometry plots of bone marrow and splenic cells.

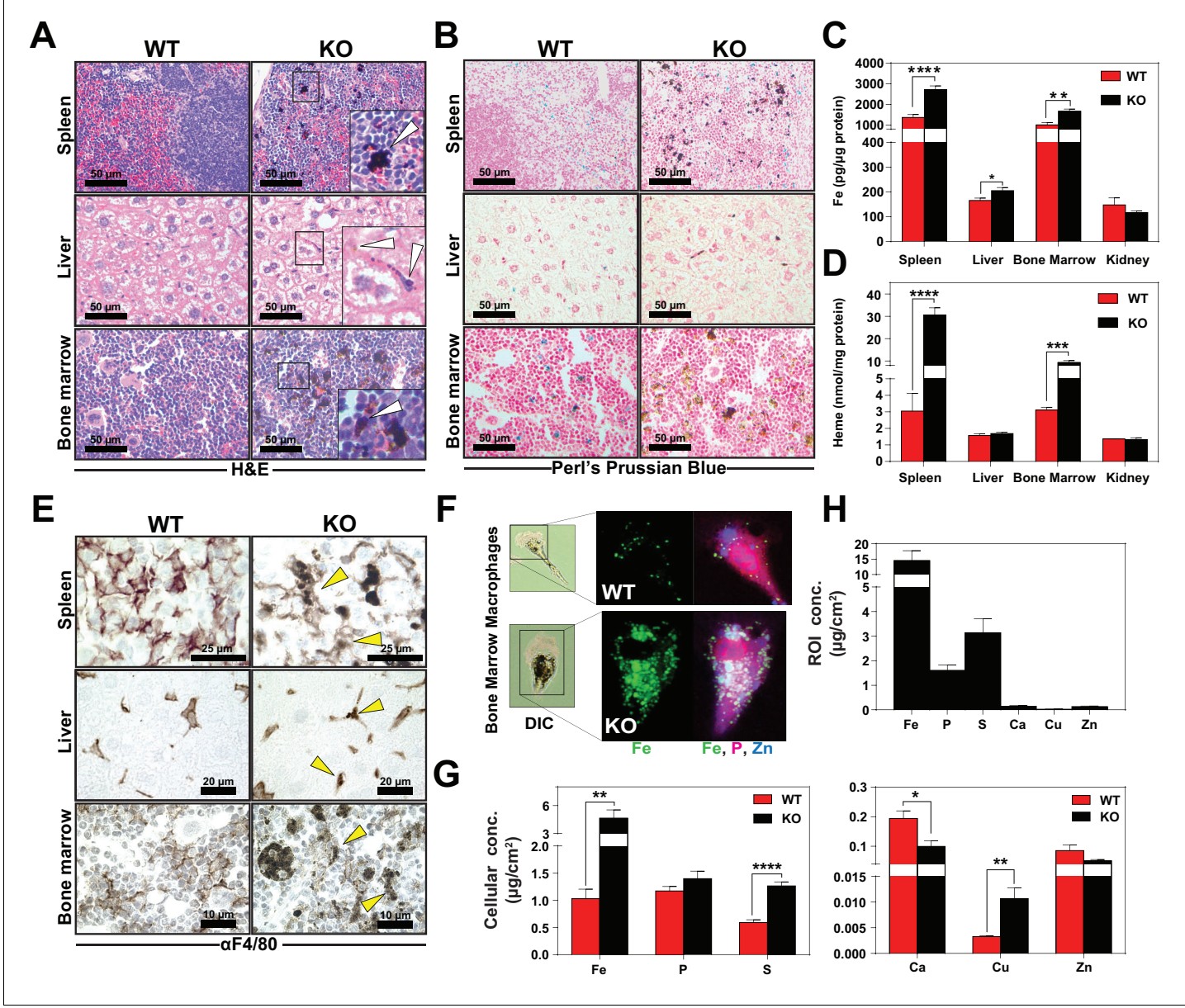

**Figure 3.** Heme accumulates within RES macrophages of KO mice Histochemical staining of spleen, liver and bone marrow tissue sections of WT and KO mice with H and E (A) or Perl's Prussian blue (B). Arrows indicate dark pigments in KO tissues. Images shown are representative of at least three mice. Quantification of tissue iron (C) and heme (D) by ICP-MS and UPLC, respectively in tissues of mice fed a standard diet (n = 6–17). (E) F4/80 immunohistochemistry analysis of paraffin-embedded tissue sections of mice. Yellow arrows indicate dark pigments within F4/80-positive cells. At least three mice were analyzed per genotype. (F) XFM of F4/80-positive bone marrow macrophages from WT and KO mice. (G) Quantification of total cellular concentrations of indicated elements, organized by cellular abundance. (H) ROI concentration within dark aggregates in KO BMMs. Quantifications were measured for n = 5 cells per genotype. *p<0.05; **p<0.01; ***p<0.001; ****p<0.0001.

The online version of this article includes the following figure supplement(s) for figure 3:

**Figure supplement 1.** Quantification of tissue metals and F4/80 IHC controls.

WT organs (*Figure 3D*, *Figure 3—figure supplement 1E*), which indicates that the higher iron content observed in these tissues is due to the accumulation of heme.

We next sought to determine the identity of the cells containing the dark pigments. Immunohistochemistry using a macrophage-specific antibody showed dark inclusions in F4/80+ cells in all three RES tissues (*Figure 3E*, right panel arrows, *Figure 3—figure supplement 1F–H*). To spatially resolve the metal content at the subcellular level, we employed synchrotron-based X-ray fluorescence

microscopy (XFM) which provides quantitative information about elemental distribution at the nano-scale. F4/80$^+$ macrophages were isolated from the bone marrow by magnetic bead separation, seeded on an appropriate sample support and visualized using XFM (*Chen et al., 2015*; *Vogt, 2003*). Overall, KO macrophages contained significantly higher concentrations of cellular iron (*Figure 3F, G*), copper, sulfur, and calcium (*Figure 3G*). However, region-of-interest (ROI) quantification around the dark pigments in KO macrophages confirmed that iron was the most abundant trace element within this region (*Figure 3H*).

## Dietary iron deficiency disrupts iron metabolism gene expression in KO mice and results in lethality

Total body iron is a composite of recycled heme-iron from damaged RBCs and dietary iron absorption. Since heme-iron accumulates within KO macrophages which facilitate iron recycling during RBC turnover, we hypothesized that KO mice might be more susceptible to a low-iron diet. The standard rodent diet contains ≈380 ppm of iron, which is significantly greater than the amount of iron (25 to 100 ppm) required to sustain normal erythropoiesis and the growth of laboratory mice (*Sorbie and Valberg, 1974*). We fed weanling WT and KO mice a low-iron diet (1.8 to 2 ppm Fe) and measured the impact of limiting dietary iron on their hematological indices. A significant percentage of KO mice, but not WT mice, died within 20 weeks on the low-iron diet (*Figure 4A*). Importantly, for the first five weeks, the hematocrits of WT and KO mice on the low-iron diet are comparable (*Mazzaccara et al., 2008*) (*Figure 4B*), but by 8 weeks the hematocrits of KO mice were significantly lower (p<0.01), and by 20 weeks they were severely anemic (13.2 ± 2.3 KO versus 39.8 ± 2.2 WT, p<0.0001; *Figure 4C*). These results imply that the defect in heme-iron recycling in KO mice becomes more pronounced after five weeks on an iron-deficient diet. We therefore focused our subsequent analyses on mice that were on low-iron diets for five weeks to detect any molecular and physiological changes in the absence of overt hematological differences. Analyses of serum from iron-deficient animals showed no significant differences in serum iron, TIBC, or transferrin saturation, but serum ferritin levels remained elevated in KO mice on a low-iron diet (Table 2 in *Supplementary file 1*, 2 ppm diet). The levels of iron and heme retained within RES organs of KO mice were also significantly higher than WT mice (*Figure 4D,E*, *Figure 4—figure supplement 1A*), suggesting that the heme-iron stores in KO mice were not bioavailable despite systemic iron-deficiency.

In response to iron deficiency, the spleens of WT mice increased by approximately 120% (*Figure 4F*) (*Lenox et al., 2005*; *Perry et al., 2009*; *Socolovsky, 2007*), with a concomitant increase (7.5-fold) in immature Ter119$^+$ cells, a hallmark of stress erythropoiesis (*Figure 4G,H*). In contrast, the stress erythropoiesis response in iron-deficient KO mice was significantly attenuated (*Figure 4F–H*, *Figure 4—figure supplement 1B*), with erythroblast differentiation blocked at an earlier stage (population I) in the bone marrow (*Figure 4I*, *Figure 4—figure supplement 1B–D*). Iron deficiency fully restored the total numbers of mature RPM population and balance in monocyte populations (*Figure 4J,K*) in KO mice to WT levels (*Figure 4—figure supplement 1E,F*). This could reflect increased viability of erythrophagocytic RPMs due to reduced heme accumulation in these cells under systemic iron deficiency.

To interrogate the responsiveness of iron and heme metabolism genes to iron deficiency in these mice, we assembled a custom qRT-PCR array comprising probes for 90 key iron/heme metabolism genes. Iron-deficient KO mice showed significant dysregulation in 38% and 6% of these mRNAs in the spleen and liver, respectively (*Figure 4L*, *Figure 4—figure supplement 1G–I*, and Table 1 in *Supplementary file 1*). Under iron-deficient conditions, the levels of 32 iron metabolism mRNAs were significantly reduced in WT spleens (*Figure 4—figure supplement 1H*, compare WT Standard vs WT 2 ppm, KO Standard vs KO 2 ppm), but remained high in KO spleens. Together, these data suggest that KO mice are unable to respond normally to iron-deficiency, especially at sites where heme-iron accumulates at high levels such as the spleen.

## Loss of *SLC48A1* produces hemozoin biocrystals within enlarged lysosomes due to impaired erythrophagocytosis

To investigate the in vivo function of *SLC48A1* in heme-iron recycling, we labeled normal RBCs with $^{59}$Fe and followed the labeled iron after injection into WT and KO mice. WT donor mice were

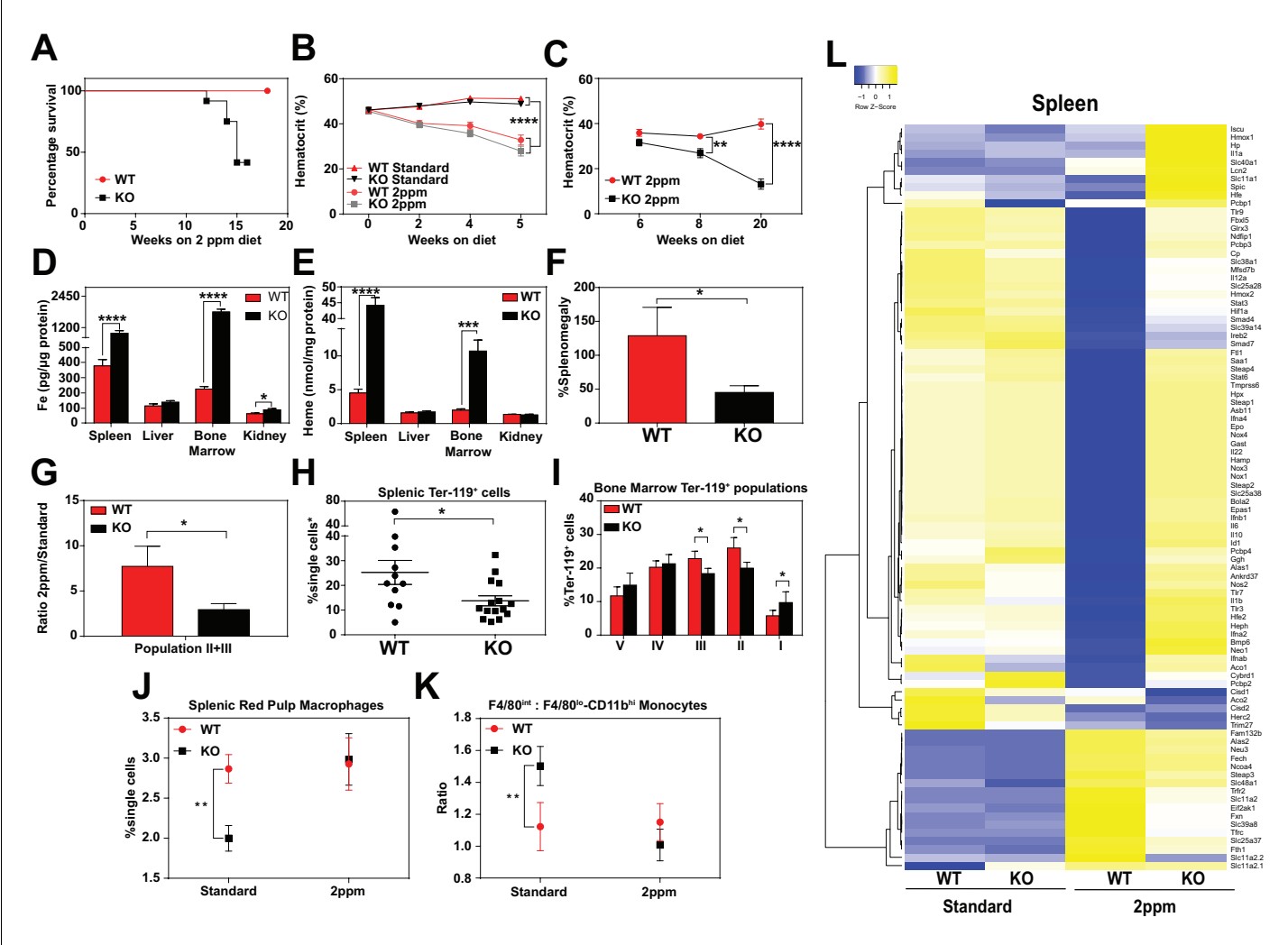

**Figure 4.** Dietary iron deficiency disrupts iron metabolism gene expression in KO mice and results in lethality. (A) Kaplan-Meier survival curve of WT and KO mice placed on a low-iron (2ppm) diet (n = 15–17, both males and females). (B–C) Hematocrits of WT and KO mice placed on a standard or low-iron (2ppm) diet. Mice were placed on respective diets supplemented with deionized water starting at 21 days of age (week 0) (n = 9–15 for 5 week data set; n = 7–11 for 20 week data set). Quantification of tissue iron (D) and heme (E) by ICP-MS and UPLC, respectively, in tissues of mice fed a low-iron (2ppm) diet (n = 6–17). (F) %Splenomegaly of WT and KO mice calculated by the percentage of increase in average wet weight of spleens between mice on low-iron versus standard iron diets (n = 9–15); (G) Ratio of 2ppm splenic Ter-119$^+$ population II+III cells to that of standard diet mice (n = 8–12); (H) Quantification of total Ter-119$^+$ cells in the spleen. The %single cells* on the y-axis denote single cells that are negative for CD4/8/41, B220 and Gr-1. Each point represents one mouse. (I) Quantification of subpopulations of Ter-119$^+$ cells represented as a percentage of total Ter-119$^+$ cells in the bone marrow (n = 7–8); (J) Quantifications of splenic RPMs in mice on a standard or low-iron (2ppm) diet, represented as a percentage of single cells analyzed (n = 9–14). (K) Quantification of the ratio of F4/80$^{hi}$ to F4/80$^{lo}$CD11b$^{hi}$ splenic monocytes from mice on a standard or low-iron (2ppm) diet (n = 9–15). At least 100,000 single cells were analyzed per sample. (L) Gene expression heat map of 90 iron metabolism genes in spleens from mice on standard or low-iron (2ppm) diet. Pearson correlation was used for comparison; average linkage (n = 9 per group, per genotype). *p<0.05; **p<0.01; ***p<0.001; ****p<0.0001.

The online version of this article includes the following figure supplement(s) for figure 4:

**Figure supplement 1.** Representative flow cytometry plots and iron metabolism gene expression levels.

treated with phenylhydrazine to induce acute hemolytic anemia and stimulate erythropoiesis, followed by an injection of $^{59}$Fe-citrate which is incorporated into heme of newly-formed RBCs. Three days later, ≈80% of $^{59}$Fe label was incorporated into heme and the $^{59}$Fe-labeled RBCs were opsonized and injected into WT and KO recipient mice (*Franken et al., 2015*; *Soe-Lin et al., 2009*) that had been maintained on a low-iron diet for six weeks (*Figure 5A*). The retention and distribution of $^{59}$Fe in the tissues of the recipient mice were monitored over a period of 96 hr. The spleens of WT

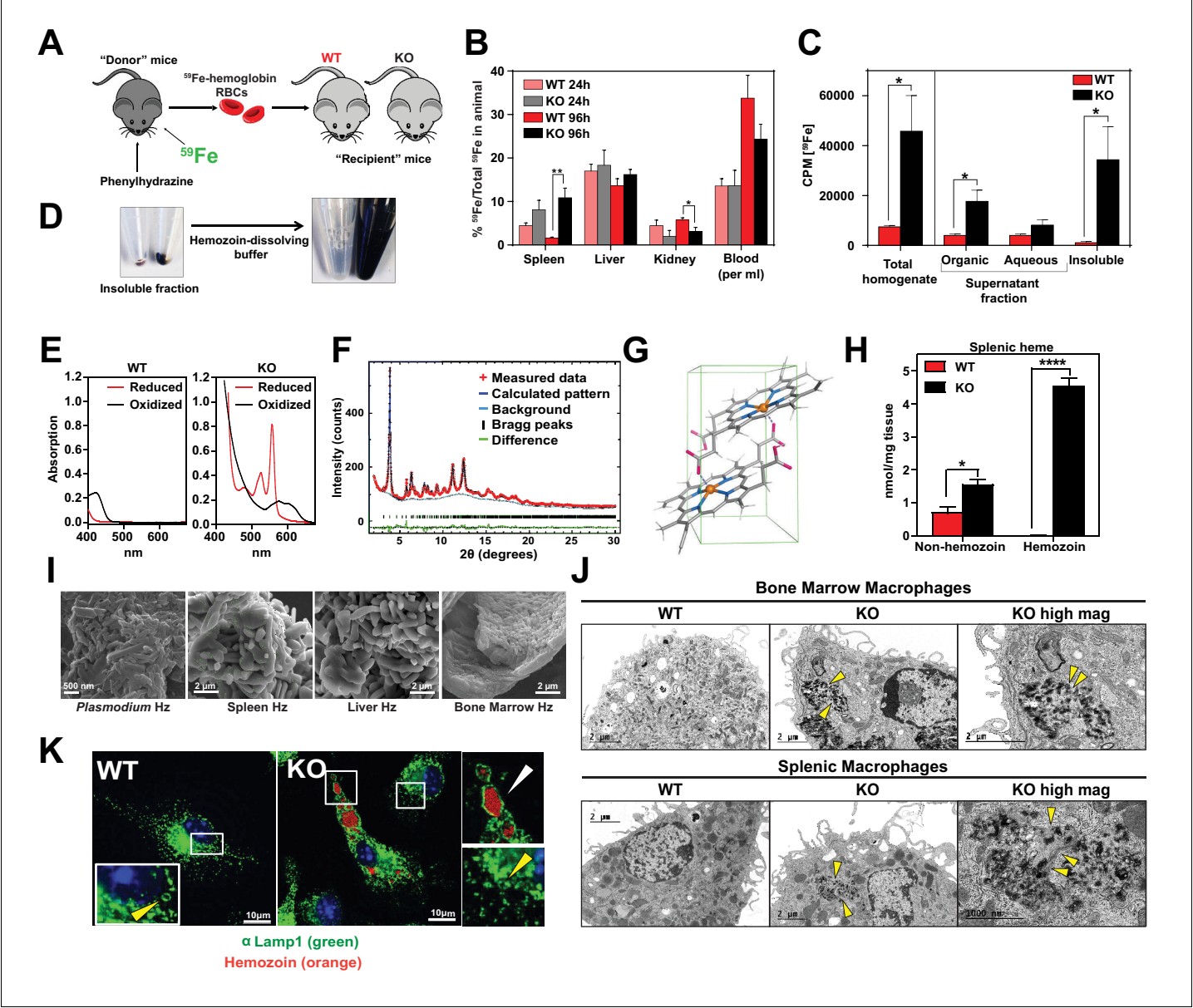

**Figure 5.** Loss of *SLC48A1* produces hemozoin biocrystals within enlarged lysosomes due to impaired erythrophagocytosis. (A) Experimental design of 59Fe labeling and in vivo recycling. (B) Quantification of 59Fe retained in tissues, represented as the ratio of the amount of radioactivity within an organ to that of the entire animal. (C) 59Fe retained in differentially extracted fractions of the spleen at 96 hr, represented as counts per min. Total homogenate: homogenized and proteinase-treated whole spleen; Organic: ethyl acetate extractable [59Fe]heme; Aqueous: ethyl acetate non-extractable 59Fe; Insoluble fraction: proteinase-insoluble fraction containing 59Fe (n = 4–6 across all groups and timepoints). (D) Image of insoluble fractions before and after dissolving in buffer containing 100 mM NaOH, 2% SDS and 3 mM EDTA. (E) Visible spectra of dissolved insoluble fractions. (F) Powder x-ray diffraction of purified insoluble fraction from KO spleens (red: measured data; dark blue: calculated pattern; light blue: background; green: structural plot) (G) Chemical structure of hemozoin from KO mice. (H) Quantification of splenic heme by spectrophotometric measurements (n = 3). (I) Scanning electron microscopy of hemozoin isolated from *Plasmodium falciparum*, KO spleen, liver, and bone marrow. (J) Transmission electron microscopy of F4/80+ bone marrow and splenic macrophages from WT and KO mice. At least 3 cells were imaged per genotype. (K) Confocal microscopy of bone marrow macrophages from WT and KO mice probed with anti-LAMP1 antibody and secondary alexa-488 antibody. Hemozoin is pseudocolored as orange. White arrow points to hemozoin-laden vesicle, yellow arrow points to non hemozoin-laden vesicle. At least 20 cells were analyzed per genotype. *p<0.05; **p<0.01.

The online version of this article includes the following figure supplement(s) for figure 5:

**Figure supplement 1.** Supporting data for hemozoin in WT and KO mice.

mice accumulated $^{59}$Fe within 24 hr followed by a decrease at 96 hr with a concomitant increase of $^{59}$Fe in the circulation and kidneys (*Figure 5B*). By contrast, the spleens of KO mice accumulated more $^{59}$Fe over the course of 96 hr with limited amounts of $^{59}$Fe in the circulation and kidneys, suggesting that the release of $^{59}$Fe from phagocytosed RBCs was impaired (*Figure 5B*).

To discriminate $^{59}$Fe from [$^{59}$Fe]heme, homogenized spleens were subjected to high-speed centrifugation to precipitate proteinase-resistant and detergent-insoluble fractions. The resulting supernatant was differentially extracted by ethyl-acetate to separate heme from iron. While KO spleens accumulated significantly greater levels of $^{59}$Fe compared to WT spleens, these differences were primarily due to $^{59}$Fe counts within the organic heme fraction ([$^{59}$Fe]heme) and not in the aqueous iron fraction ($^{59}$Fe) (*Figure 5B,C*).

A large portion of $^{59}$Fe was detected in the detergent-insoluble fraction of KO spleens (*Figure 5C*) and livers (*Figure 5—figure supplement 1A*). A proteinase and detergent resistant fraction enriched in iron is characteristic of hemozoin, a crystallized form of heme generated during the digestion of hemoglobin by blood-feeding organisms such as *Plasmodium* (*Egan, 2008*; *Francis et al., 1997*). To determine whether the insoluble fractions from KO tissues had hemozoin-like properties, we treated the fractions with buffers known to dissolve hemozoin (*Figure 5D*). Spectroscopic analysis of the dissolved detergent-resistant fraction revealed absorption spectra characteristic of heme (*Figure 5E*, *Figure 5—figure supplement 1B*). High resolution X-ray powder diffraction of the purified insoluble fraction from the KO spleens confirmed that the dark pigment is identical to malarial hemozoin (*Figure 5F*) (*Coronado et al., 2014*; *Slater et al., 1991*). The structure reveals a repeating unit of heme dimers linked together by iron-carboxylate bonds to one propionic acid side chain with the adjacent heme dimers forming stable chains by hydrogen bonding via the free propionate side chain (*Figure 5G*, *Figure 5—figure supplement 1C*) (*Coronado et al., 2014*). Spectrophotometric analysis of the dissolved hemozoin fractions showed substantial amounts of hemozoin-heme present in the spleens and livers of KO mice, but not in WT mice (*Figure 5H*, *Figure 5—figure supplement 1D*). Notably, a significant amount of heme in non-hemozoin form was also found in KO spleens (2-fold of WT; *Figure 5H*).

Scanning electron microscopy images showed mammalian hemozoin was heterogeneous; purified hemozoin crystals from spleen and liver appeared larger than *Plasmodium falciparum* hemozoin, while bone marrow-purified hemozoin was similar in size (*Figure 5I*). Transmission electron microscopy with purified F4/80$^+$ tissue-resident macrophages from the spleen and bone marrow showed that hemozoin was sequestered within a membrane-enclosed compartment (*Figure 5J*). However, isolated KO monocytes that were differentiated into macrophages in vitro did not contain any hemozoin (*Figure 5—figure supplement 1E*), indicating that hemozoin formation occurs in vivo after tissue resident macrophages are established. Confocal microscopy using antibodies against LAMP1, which has been previously demonstrated to be a strong marker of erythrophagosomes in macrophages (*Delaby et al., 2012*; *Huynh et al., 2007*), showed that all of the hemozoin in isolated-F4/80$^+$ macrophages from KO bone marrow is enclosed within LAMP1-positive compartments (*Figure 5K*, *Figure 5—figure supplement 1F*). Furthermore, these hemozoin-containing LAMP1-positive vesicles were significantly larger (diameter 570 to 4300 nm) compared to typical lysosomes which range from 50 nm to 500 nm (*Bandyopadhyay et al., 2014*) (*Figure 5K*, compare vesicles with white vs yellow arrows; *Figure 5—figure supplement 1G*).

Intracellular hemozoin is typically known to be inert and ingested malarial hemozoin can remain unmodified for long periods of time within human monocytes (*Schwarzer et al., 1993*). To test whether mouse hemozoin could be used as a source of iron when macrophages are forced to undergo apoptosis, we administered clodronate liposomes to iron-deficient WT and KO mice. RES macrophages were completely depleted by day three post-injection but KO spleens, livers and bone marrows still retained hemozoin, which were now extruded into the interstitial space (not shown). However, this hemozoin was not bioavailable as an iron source to correct iron deficiency anemia even after seven days (*Figure 5—figure supplement 1H*). Altogether, these results strongly indicate that in the absence of *SLC48A1*, RES macrophages are unable to recycle and degrade heme from ingested RBC resulting in heme accumulation and biocrystallization into hemozoin within enlarged phagolysosomes.

## *SLC48A1* deficiency confers cellular heme tolerance during erythrophagocytosis

To evaluate the impact of SLC48A1 deficiency at the cellular level, we analyzed the ability of bone marrow-derived macrophages (BMDMs) to recycle heme-iron derived from phagocytosed RBCs. Although WT and KO BMDMs engulfed similar numbers of opsonized RBCs (*Figure 6—figure supplement 1A*), SLC48A1-deficient cells accumulated significantly more heme even after 72 hr post-EP (*Figure 6A*). As heme is cytotoxic and damaging to macrophages (*Kovtunovych et al., 2010*), we measured cellular markers of oxidative stress during the two critical phases of heme-iron recycling after EP - the early (4 hr) and late (24 hr) phase. Reactive oxygen species (ROS) production (*Figure 6B*) (*Delaby et al., 2005*; *Kovtunovych et al., 2010*) and the ratio of oxidized to reduced glutathione levels (GSSG:GSH) (*Figure 6C*) was significantly lower in KO cells during the early phase, when active digestion of red cells takes place. Consistent with this early suppression of ROS, lactate dehydrogenase (LDH) release into the growth medium (*Rayamajhi et al., 2013*) was also significantly reduced in KO cells in the late phase (*Figure 6D*), with no loss in cell adherence (*Figure 6—figure supplement 1B*) (*Delaby et al., 2005*). Together, these results imply that SLC48A1 deficiency protects the macrophages from the damaging effects of EP (*Delaby et al., 2005*).

## Haploinsufficiency of *HMOX1* in *SLC48A1*-deficient animals causes perinatal lethality

The in vivo and cell biological studies posit that loss of SLC48A1 confers heme tolerance by confining heme to the phagolysosome and preventing its degradation by HMOX1. However, EP studies reveal that HMOX1 induction and abundance were comparable between WT and KO BMDMs over 72 hr post-EP (*Figure 6E*) These results raised the possibility that, in the absence of SLC48A1, an alternate albeit less-efficient heme transport pathway could translocate heme from the phagolysosome for degradation by HMOX1. To evaluate the genetic contribution of HMOX1, we generated HMOX1 HET and DKO (HMOX1 KO; SLC48A1 KO double knockout) double mutant mice. Analysis of offspring from HMOX1 HET and HMOX1 HET; SLC48A1 KO intercrosses showed reduced numbers of viable HMOX1 KO progeny (Observed/Expected: HMOX1 KO 9/64 versus DKO 9/86) regardless of the presence of *SLC48A1* (*Figure 6F*, *Figure 6—figure supplement 1C*). Consistent with this observation, HMOX1 KO and DKO mice showed similar reductions in RES macrophages and erythropoietic profiles from the bone marrow and spleen at ten weeks of age (*Figure 6—figure supplement 1D–I*), and developed anemia as they grew older (not shown).

Unexpectedly, heterozygous offspring from HMOX1 HET; SLC48A1 KO intercrosses did not show Mendelian ratio at birth, with almost 40% embryonic lethality (Observed/Expected: HMOX1 HET 133/128 versus HMOX1 HET; SLC48A1 KO 108/172) ($p < 0.001$ by chi-square test) (*Figure 6F*, *Figure 6—figure supplement 1C*). Flow cytometry analyses of HMOX1 HET; SLC48A1 KO mice showed an enhanced reduction in splenic RPMs (*Figure 6G*) and total Ter-119[+] cells in the bone marrow (*Figure 6H*) compared to HMOX1 HET mice. The bone marrow subpopulations of Ter-119[+] cells were also significantly reduced in populations II to V in HMOX1 HET; SLC48A1 KO mice (*Figure 6I*). The spleens of HMOX1 HET; SLC48A1 KO mice had fewer Ter-119[+] cells with more immature Ter-119[+] cells (*Figure 6J,K*), suggesting ineffective stress erythropoiesis. Together these results reveal that inhibition of HMOX1 in the absence of SLC48A1 has two different outcomes depending on the degree of HMOX1 impairment. Partial reduction or haploinsufficiency of *HMOX1*, as observed in HMOX1 HET; SLC48A1 KO mice, increases perinatal lethality and a shift to synthetic lethality. However, complete inhibition of *HMOX1* leads to loss in cell viability (*Figure 6—figure supplement 1J, K*) and greater embryonic mortality of DKO and HMOX1 KO mice (*Figure 6—figure supplement 1C*), demonstrating a delicate balance between heme transport and degradation.

## Discussion

Our results show that (SLC48A1) KO mice are defective in transporting heme across the phagosomal membrane during EP, resulting in large amounts of heme accumulation and hemozoin formation within phagolysosomes of RES macrophages. Heme retention leads to impaired iron recycling for erythrocyte production in the bone marrow, stimulating extramedullary erythropoiesis similar to the effects caused by iron-deficiency (*Figure 6L*). However, the overall impact on RBC production and

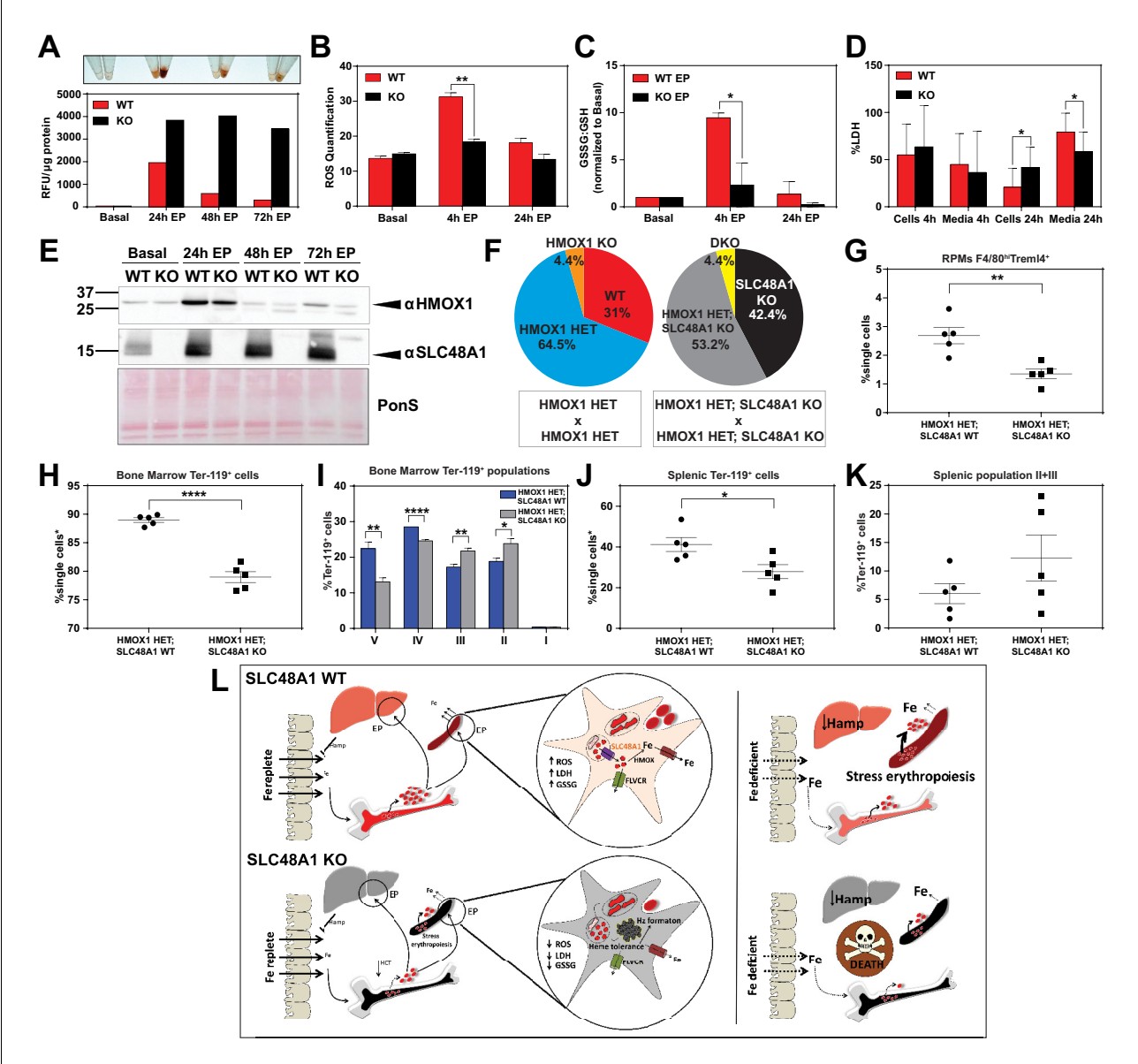

**Figure 6.** *SLC48A1* deficiency confers cellular heme tolerance during erythrophagocytosis and haploinsufficiency of *HMOX1* in *SLC48A1*-deficient animals causes perinatal lethality. (**A**) Images of cell lysates and quantification of heme content in WT and KO BMDMs at basal (no treatment), 24 hr, 48 hr and 72 hr post-EP (erythrophagocytosis). Results are representative of at least two experiments. (**B**) Quantification of intracellular reactive oxygen species (ROS) in WT and KO BMDMs at basal, 4 hr and 24 hr post-EP. (**C**) Ratio of cellular GSSG:GSH in WT and KO BMDMs at basal, 4 hr and 24 hr post-EP. Values for 4 hr and 24 hr are normalized to basal values. The values shown are a combination of two representative biological replicates. (**D**) LDH content in cells and media of WT and KO BMDMs at 4 hr and 24 hr post-EP, represented as percentage of total LDH in cells and media. (**E**) Immunoblots of HMOX1 and SLC48A1 protein of BMDM lysates shown in (**A**). (**F**) Percentages of pups born of the indicated genotypes for HMOX1 HET intercrosses, either on a WT or KO background. (**G**) Quantification of splenic RPMs in HMOX1 HET and HMOX1 HET; SLC48A1 KO mice by flow cytometry. (**H**) Quantification of total Ter-119$^+$ cells in the bone marrow. The %single cells* on the y-axis denote single cells that are negative for CD4/8/41, B220 and Gr-1. (**I**) Quantification of subpopulations of Ter-119$^+$ cells represented as a percentage of total Ter-119$^+$ cells in the bone marrow. (**J**) Quantification of total Ter-119$^+$ cells in the spleen. The %single cells* on the y-axis denote single cells that are negative for CD4/8/41, B220 and Gr-1. (**K**) Quantification of populations II and III of Ter-119$^+$ cells represented as a percentage of total Ter-119$^+$ cells in the spleen. At least 100,000 single cells were analyzed per sample. Each dot represents one mouse. Both male and female mice were analyzed in the experiments conducted in *Figure 6*. (**L**) Proposed model for the in vivo and in vitro function of *SLC48A1* in heme-iron recycling of RBCs under systemic iron replete and deficient conditions. *p<0.05; **p<0.01; ****p<0.0001.

The online version of this article includes the following figure supplement(s) for figure 6:

**Figure supplement 1.** Extended characterization of HMOX1; SLC48A1 mice.

anemia is modest on a standard iron diet (380 ppm). These findings support the conclusion that mice on an iron-rich diet can circumvent a block in heme-iron recycling by relying more heavily on dietary iron for erythropoiesis (*Ganz and Nemeth, 2006*). This is in stark contrast to humans where the daily iron absorption does not exceed 10% of the amount of recycled iron utilized for erythropoiesis (*Kautz and Nemeth, 2014*; *Muñoz et al., 2009*; *Winter et al., 2014*). However, when KO mice are fed a low-iron diet, the block in heme-iron recycling is exacerbated. Furthermore, when KO mice encounter prolonged dietary iron limitation (8 to 20 weeks), they are unable to sustain erythropoiesis, implying that it takes approximately eight weeks for body iron stores to be depleted in the absence of heme-iron recycling. Indeed, results from the *in vivo* [59]Fe-labeled RBC studies show that the relative proportion of [59]Fe retained within the spleens of KO mice are 50.3% hemozoin: 34.4% heme: 15.3% non-heme iron after 96 hr (*Figure 5C*). Assuming that ineffective heme-iron recycling reduces recycled iron according to the law of exponential decay (*Leike, 2002*) and RBCs have a lifespan of ~40 days, this [59]Fe distribution pattern would predict that KO mice would become significantly anemic after six weeks as hemozoin accumulates after RBC turnover. Another potential contributing factor is the mouse genetic background. The KO mice were developed on a mixed genetic background (75% C57BL/6J, 25% 129/SvJ). Published studies comparing the iron status of C57BL/6J and 129/SvJ mice on an iron-balanced diet have shown that 129/SvJ mice have more hepatic iron stores (*Dupic et al., 2002*; *Wang et al., 2007*). We recently generated KO mice on a pure C57BL/6J background and placed these mice on an iron-deficient diet post-weaning (P21). KO C57BL/6J mice showed severe anemia within 5.5 weeks compared to WT littermate controls, which approximates one lifespan of circulating erythrocytes (not shown). This result indicates that genetic strain differences that influence body iron stores are important modifiers of the SLC48A1 mutant phenotype. Taken together, we conclude that KO mice reach a 'metabolic threshold' on a low-iron diet, that is a physiological transition from an intermediate iron state to severe iron limitation, which eventually lead to mortality in these animals (*Figure 6L*, right panel).

Even though WT and KO mice had similar hematocrits at five weeks on a low-iron diet, KO mice were unable to undergo stress erythropoiesis and showed dysregulation of over 35 iron and heme metabolism genes in their spleens and livers. Typically under iron-deficient conditions, the production of the liver hormone hepcidin is suppressed, which results in greater mobilization of iron by SLC40A1 to iron-deprived compartments (*Ganz and Nemeth, 2006*; *Nemeth et al., 2004*). The inability to sustain erythropoiesis, despite a significant suppression in hepatic hepcidin expression in KO iron-deficient mice (*Figure 4—figure supplement 1G*), suggest that these mice are iron-deprived even though the RES tissues are heme-iron loaded, that is systemic iron status is uncoupled from tissue and cellular iron status in the absence of *SLC48A1*. While hepatic hepcidin is considered to be the major facilitator of systemic iron homeostasis, recent studies have uncovered an autocrine role for locally-produced hepcidin in regulating cardiac iron homeostasis (*Lakhal-Littleton et al., 2016*). Indeed, KO mice show elevated splenic hepcidin regardless of iron status (*Figure 4—figure supplement 1H*), implying a role for splenic hepcidin in stress erythropoiesis.

In the absence of *SLC48A1*, heme accumulates in the erythrophagosome of RES macrophages in the form of hemozoin biocrystals. Heme-iron recycling of senescent erythrocytes occurs in the RES macrophages - mainly in the splenic RPMs and, to some extent, in liver Kupffer cells and bone marrow macrophages (*Beaumont and Delaby, 2009*; *Ganz, 2012*; *Theurl et al., 2016*). Our results establish a direct role for SLC48A1 in transporting heme from the phagolysosomal compartments to the cytosol for degradation by HMOX1. Unlike (SLC48A1) KO mice, loss of *HMOX1* causes over 90% embryonic lethality. Why do (SLC48A1) KO mice survive while HMOX1 KO mice show high embryonic mortality? Our results show that in the absence of *SLC48A1*, high concentrations of heme is sequestered as hemozoin within the acidic phagolysosome, resulting in cellular heme tolerance. By contrast, in the absence of *HMOX1*, heme accumulates within the cytosol resulting in cytotoxicity and embryonic lethality. Genetic epistasis would have predicted that the heme sequestration would rescue the embryonic lethality of the DKO mice. Possible explanations for this unexpected finding could be that *HMOX1* has additional functions beyond heme degradation, or that the heme degradation products are required for cell differentiation and specification in vivo. Our results support both of these explanations and further reveal that KO mice require a fully-operational heme degradation pathway to confer complete heme tolerance, as partial reduction of *HMOX1* in the absence of *SLC48A1* show significant embryonic lethality with severe impairment in macrophage and erythroblast maturation in the bone marrow and spleen (*Figure 6F–K*) raising the possibility that functional

HMOX1 requires SLC48A1 on the erythrophagosomal membranes. In the absence of SLC48A1, bioactive HMOX1 drops below the 50% expected in HMOX1 HET animals.

Blood-feeding parasites degrade hemoglobin in their digestive vacuole and accumulate heme in the form of hemozoin (*Chen et al., 2001*; *Toh et al., 2010*). Although the exact mechanisms of in vivo hemozoin formation are not well understood, it is generally agreed upon that hemozoin formation is a heme detoxification method used to protect the parasite from heme toxicity (*Hempelmann, 2007*; *Toh et al., 2010*). Hemozoin biocrystals are thought to be well-tolerated by both the parasite and subsequent host cells which ingest it, despite some reports on the inflammatory responses towards hemozoin ingestion (*Basilico et al., 2003*; *Schwarzer et al., 1993*). We found hemozoin within LAMP1-positive intracellular vesicles, which are typically between pH 4 and 6 (*Johnson et al., 2016*; *Mellman et al., 1986*). Acidic pH appears to be a common requirement for hemozoin formation in parasitic digestive vacuoles, in vitro and in macrophages (*Egan et al., 2001*; *Shio et al., 2010*). To our knowledge, hemozoin formation in mammals has not been documented prior to this study. This phenomenon observed in KO spleens, livers and bone marrows suggests that macrophages are able to elicit a protective mechanism against high concentrations of heme by exploiting the low pH environment of lysosomes. It is important to note that KO mice have fewer RPMs, which implies that hemozoin sequestration may be incomplete or insufficient to confer complete heme tolerance, or that hemozoin itself could cause detrimental effects to these cells. Hemozoin in KO RES macrophages leads to tissue heme accumulation in these mice without causing major damages to tissue architecture. We speculate that pharmacologic inhibition of SLC48A1 in humans would lead to tolerance and protection from heme toxicity and iron overload.

# Materials and methods

**Key resources table**

| Reagent type (species) or resource | Designation | Source or reference | Identifiers | Additional information |
|---|---|---|---|---|
| Strain, strain background (*Mus musculus*, 129/SvJ/C57BL/6J) | SLC48A1, HMOX1 | This paper, Materials and methods subsection animals | | Mouse strain |
| Biological sample (*Mus musculus*) | Primary bone marrow-derived macrophages | SLC48A1/ HMOX1 mice | | |
| Biological sample (*Plasmodium falciparum*) | Hemozoin | *Plasmodium falciparum* | | Gift from Dr. Paul Sigala |
| Antibody | SLC48A1 (Rabbit, polyclonal) | PMID: 30248094 | | 1:500 |
| Antibody | F4/80 (Rat, polyclonal) | Invitrogen | MF48000 RRID:AB_10376289 | 1:1000 |
| Antibody | HMOX1 (rabbit, polyclonal) | Enzo | ADI-SPA-896 RRID:AB_10614948 | 1:1000 |
| Antibody | LAMP1 (rat, polyclonal) | Developmental Studies Hybridoma Bank | 1D4B RRID:AB_2134500 | 1:100 |
| Antibody | Anti-F4/80 microbeads | Miltenyi Biotech | 130-110-443 | Beads |
| Recombinant DNA reagent | guide RNA | Sage Laboratories | | TAGGGACGGTGGTC TACCGACAACCGG |
| Recombinant DNA reagent | Cas9 RNA | Trilink Biotechnologies | | |
| Sequence-based reagent | HMOX1 KO F | PMID: 24963040 | | GCTTGGGTGGA GAGGCTATTC |

*Continued on next page*

*Continued*

| Reagent type (species) or resource | Designation | Source or reference | Identifiers | Additional information |
|---|---|---|---|---|
| Sequence-based reagent | HMOX1 KO R | PMID: 24963040 | | CAAGGTGAGATGACAGGAGATC |
| Sequence-based reagent | HMOX1 WT F | PMID: 24963040 | | GTACACTGACTGTGGGTGGGGGAG |
| Sequence-based reagent | HMOX1 WT R | PMID: 24963040 | | AGGGCCGAGTAGATATGGTAC |
| Sequence-based reagent | Custom qPCR array | Qiagen; this paper | CLAM25204D | |
| Commercial assay or kit | Stanbio Iron and TIBC kit | VWR | 10152–550 | |
| Commercial assay or kit | mouse ferritin ELISA kit | Abcam | ab157713 | |
| Commercial assay or kit | LDH kit | Sigma | TOX7 | |
| Commercial assay or kit | ROS kit | Sigma | MAK142 | |
| Commercial assay or kit | GSH assay kit | Abcam | ab138881 | |
| Chemical compound, drug | Clodronate liposomes | Clodronate liposomes | C-005 | |
| Software, algorithm | Heatmapper software | PMID: 27190236 | | |
| Software, algorithm | PRISM seven software | Graphpad | | |
| Other | $^{59}$FeCl3 | Perkin Elmer | NEZ037001MC | Radioactive material |

## Animals

All mice used were housed in a 12 hr light-dark cycle. For *SLC48A1* pups, genetic segregation was computed on 21 day old (P21) mice pups. *SLC48A1* mice were genotyped from tail genomic DNA extracts using a custom ordered TaqMan SNP Genotyping Assays probe (ThermoFisher Scientific) on a Bio-rad CFX Connect system. For *HMOX1/SLC48A1* mice, genetic segregation was computed on 5 day old (P6) mice pups via toe-clip DNA extracts. *HMOX1* was genotyped by PCR using primers HMOX1 KO Forward 5'-GCTTGGGTGGAGAGGCTATTC-3', HMOX1 KO Reverse 5'-CAAGGTGAGATGACAGGAGATC-3', HMOX1 WT Forward 5'-GTACACTGACTGTGGGTGGGGGAG-3', HMOX1 WT Reverse 5'-AGGGCCGAGTAGATATGGTAC-3'. Mice in all studies were males unless otherwise noted, although initial experiments to exclude gender variation were done using both males and females. All animal protocols were approved by the Institutional Animal Care and Use Committee at the University of Maryland, College Park (IACUC Animal Study Protocol R-NOV-18–61).

## Generation of KO mice

Guide and Cas9 RNAs: The guide RNA (5'-TAGGGACGGTGGTCTACCGACAACCGG-3') was purchased from Sage Laboratories 2033 Westport Center Drive, St Louis, MO. Cas 9 RNA was purchased from Trilink Biotechnologies, San Diego, CA. The guide RNA and Cas9 RNA were combined at a concentration of 5 ng/μl (each) in 10 mM Tris, 0.25 mM EDTA (pH 7.5) for injection.

Pronuclear Injection: Pronuclear injection was performed using standard procedures (*Behringer et al., 2014*). Briefly, fertilized eggs were collected from superovulated C57BL/6J × 129/SvJ F$_1$ females approximately 9 hr after mating with C57BL/6J male mice. Pronuclei were injected with a capillary needle with a 1–2 μm opening pulled with a Sutter P-1000 micropipette puller. The RNAs were injected using a FemtoJet 4i (Eppendorf) with continuous flow estimated to deposit approximately 2 pl of solution. Injected eggs were surgically transferred to pseudo-pregnant CB6 F$_1$ recipient females.

Genotyping: DNA was obtained from founder ($F_0$) animals by tail biopsy, amplified by PCR (Forward 5'-TGCACCTGTGACTCGGCG-3' Reverse 5'-TAGGTCCCGCCACGTTCATAA-3') and sequenced to determine the mutation. $F_0$ animals carrying mutations were crossed to C57BL/6J animals and the resulting heterozygous $F_1$ animals were either intercrossed (F) to generate homozygous mutant animals or back crossed (N) to C57BL/6J mice for propagation. All mice used were $N_1F_2$ generation derived from intercrosses between heterozygotes, and speed congenics (DartMouse, NH) determined the genetic background to be about 75% C57BL/6J and 25% 129/SvJ. WT mice denote wildtype littermates of KO mice.

## Dietary study

WT and KO mice obtained from SLC48A1 HET crosses were weaned at 21 days of age (P21) and placed on their respective diets, supplemented with deionized water. Standard rodent diet was obtained from Envigo and the two ppm (TD.09127) Fe diet was custom ordered from Envigo, Madison, WI. Body weights and feed intake per cage were measured weekly to ensure the conditions of the animals. Blood was collected by retro-orbital bleeding using microcapillary tubes (Fisher Scientific, cat. number 22–362566). At the end of five weeks, mice were sacrificed by cardiac perfusion using Dulbecco's phosphate-buffered saline (DPBS) (Gibco, cat. number 14190250) under anesthesia (10% ketamine, 8% xylazine mix). Prior to perfusion, whole blood was collected into tubes and allowed to clot at room temperature for 45 min and serum was separated from the sample by centrifugation at 2000 g for 10 min. For the prolonged dietary iron study where the Kaplan Meier survival curve was obtained, whole blood hematocrits were obtained every two weeks until week 14. All surviving KO mice were sacrificed when the survival rate fell below 50% (~week 16), while WT mice were kept on the diets until week 18.

## Immunohistochemistry and histology

Paraffin-embedded tissue sections were processed for antigen retrieval by heat-induced epitope retrieval in citrate buffer pH 6 (DAKO, Glostrup, Denmark, cat. number S2367). After epitope retrieval, sections were then incubated with either rat anti-F4/80 (Invitrogen, cat. number MF48000) (1:1000 in blocking buffer TBS with 2% FBS) or rabbit anti-SLC48A1 (*Zhang et al., 2018*) (1:500) overnight at 4°C. Polyclonal SLC48A1 antibody serum was generated in rabbit using the C-terminal 17 amino acid peptide sequence (YAHRYRADFADIILSDF) of human SLC48A1 as antigen (Epitomics, Inc). Sections were then incubated with secondary biotinylated anti-rat antibody (Vector labs, cat. number BA-9400) for 30 min at room temperature. Signals were detected by DAB substrate or the alkaline phosphatase red substrate (Vector labs, cat. number: SK-5100) incubation and slides were lightly counterstained with hematoxylin. H and E and Perl's Prussian blue stainings were conducted by Histoserve, Inc.

## Crude membrane prep and western blots for SLC48A1

Tissues were snap frozen in liquid nitrogen and ground using a ceramic pestle and mortar maintained ice-cold. In a dounce homogenizer, powdered tissue samples were added to membrane prep buffer (250 mM Sucrose, 1 mM EDTA, 10 mM Tris-HCl pH 7.4, 3X protease inhibitor cocktail). Samples were dounce homogenized until no obvious chunks of tissue were observed and the number of strokes used for homogenization was kept consistent across all samples. Homogenates were centrifuged at 800 g for 10 min at 4°C. The supernatant was transferred to ultracentrifugation tubes and spun at 100,000 g for 2 hr at 4°C. The pellet from ultracentrifugation was then resuspended in lysis buffer (150 mM NaCl, 1 mM EDTA, 20 mM HEPES pH 7.4, 2% Triton-X, 3X protease inhibitor) and sonicated to ensure complete lysis. The sample was then centrifuged at 11,000 g for 30 min at 4°C and the insoluble pelleted debris was discarded. Protein concentration of the supernatant was determined using the BCA assay (Pierce BCA Protein Assay Kit, ThermoFisher Scientific, cat. number 23225). Samples were mixed with SDS-loading buffer without heating and electrophoretically separated on a 4–20% Criterion TGX Precast Midi Protein Gel (Bio-rad, cat. number 5671094) and transferred to nitrocellulose membrane. Proteins were cross-linked to membranes by UV treatment and stained by ponceau S before incubation in blocking buffer (5% nonfat dry milk in 0.05% Tris-buffered saline-Tween 20, TBS-T) for 1 hr at room temperature. Blots were then incubated overnight at 4°C in blocking buffer containing rabbit anti-SLC48A1 antibody (1:300 dilution). After three washes in TBS-

T, blots were incubated 1 hr with horseradish peroxidase (HRP)-conjugated goat anti-rabbit IgG secondary antibody (1:20000; Invitrogen cat. Number 31460) in blocking buffer. Blots were then washed five times with TBS-T and signals were visualized by using enhanced chemiluminescence (SuperSignal West Pico, Pierce) and detected using ChemiDoc Imaging Systems (Bio-rad).

## RNA extraction and quantitative Reverse-Transcriptase PCR array

Total RNA was isolated from samples using TRIzol Reagent (ThermoFisher Scientific). cDNA synthesis was done using RT2 First Strand Kit (Qiagen). The Qiagen iron metabolism RT2 profiler array was custom built (Cat. number CLAM25204D) and used with RT2 SYBR Green Fluor qPCR mastermix (Qiagen) on a CFX Connect system (Bio-rad). Analysis of gene expression data was conducted using the online data analysis web portal provided by Qiagen. Briefly, Ct values for each gene in each group (eg. WT, Standard diet) were obtained by taking the average across all mice (n = 9 per group). ΔCt values were then obtained by the following formula: ΔCt = Ct (target gene) – Ct (housekeeping gene). The housekeeping gene used was *RPL13a*. Gene expression was then calculated by the formula $2^{-(-\Delta Ct)}$. *p* values for gene expression were calculated using ΔCt values, the corresponding standard deviations and n = 9. The gene expression heatmap was generated by the Heatmapper software and clustering was performed using Pearson's distance measurement method with average clustering (*Babicki et al., 2016*).

## Hemozoin extraction and quantification

Hemozoin extraction was performed according to the method of *Deroost et al. (2012)*. Approximately 50 to 100 mg perfused mouse liver or spleen were ground in liquid nitrogen with mortar and pestle. The ground powder was resuspended in five to ten volumes of homogenization buffer (50 mM Tris/HCl pH 8.0, 5 mM $CaCl_2$, 50 mM NaCl, 1% Triton X-100% and 1% Proteinase K) and incubated overnight at 37°C with gentle shaking. For mice younger than three weeks, a whole liver or spleen was homogenized in minimum five volumes of homogenization buffer using FastPrep-24 (MP Bio) for 30 s at the 6.5 m/s setting, followed by overnight incubation at 37°C with gentle shaking. The proteinase K digested homogenate was then sonicated (Heat Systems-Ultrasonics, W350) for 1 min (20 W, pulse 1 s) and centrifuged at 11,000 g for 45 min. The supernatant was collected in a new tube (Supernatant fraction) and the pellet was washed three times in washing buffer (100 mM $NaHCO_3$, pH 9.0% and 2% SDS) with 1 min sonication and 30 min centrifugation at 11,000 g. All the supernatant from three wash steps was collected and combined (Washing fraction). The remaining pellet (Pellet fraction, Hz) was dissolved and sonicated for 1 min in dissolving buffer (100 mM NaOH, 2% SDS and 3 mM EDTA) and centrifuged at 11,000 g for 30 min to discard any insolubles. For X-ray powder diffraction analysis, after the third wash, extracted Hz was washed five more times in distilled $H_2O$ to remove the salts and detergents.

Heme concentrations of all three fractions (Supernatant, Washing and Pellet) were determined by the pyridine hemochromogen spectra method (*Barr and Guo, 2015*). A pyridine reagent mix was prepared by adding 3 mL 1 M NaOH and 6 mL pyridine to 19 mL $H_2O$ in a glass container. For oxidized spectrum, 35 μL sample and 17 μL 15 mM $K_3Fe (CN)_6$ was mixed with 1 mL pyridine reagent in a cuvette and the spectrum at 400–675 nm was recorded (Shimadzu, UV-1601). Two to five mg of powdered $Na_2S_2O_4$ were added to the mixture and the reduced spectrum was recorded at 400–675 nm. Heme concentrations were calculated by subtracting the absorbance readings at 541, 556 and 575 nm in the oxidized spectrum from the corresponding readings in the reduced spectrum to get ΔA540, ΔA556 and ΔA575, using the extinction coefficients 20.7/mM for ΔA540 ((ΔA556 - ΔA540)/20.7) and 32.4/mM for ΔA575 ((ΔA556-ΔA575)/32.4), multiplying by the dilution factor of the sample (30.06 or (1000 + 35 + 17)/35) and averaging the two results. Total heme (nmol/mg tissue) in each fraction was calculated by multiplying heme concentration with corresponding fraction volume, and then divided by the weight of homogenized tissue. Total heme (nmol/organ) in each fraction was calculated by multiplying total heme (nmol/mg tissue) with the total weight of corresponding organ. Total heme in the Supernatant and Washing fractions was summed up as non-Hz heme.

## Inductively coupled plasma mass spectrometry (ICP-MS)

Prior to metal and heme analyses, frozen tissues were added to three volumes of pure water and homogenized in ceramic bead tubes (Qiagen) using an Omni Bead Ruptor 24. For metal analysis,

homogenate aliquots were digested overnight in 5:1 $HNO_3:H_2O_2$, dried, and resuspended in 2% $HNO_3$ for analysis using an Agilent 7900 ICP-MS. Calibration standard solutions for determination of Fe, Zn, Cu and Mn were prepared from Agilent multi-element calibration standard-2A. Protein concentrations of homogenates were determined by BCA Protein Assay (Thermo Fisher Scientific) for normalization.

## Porphyrin extraction

Enough tissue homogenate preparation for ICP-MS was adjusted with water to make 50 µL of 5 mg/mL protein. The resulting suspension was extracted with 200 µL of EA (a mixture of four volumes of ethyl acetate to one volume glacial acetic acid), centrifuged at 13.5 K rpm for 0.5 min, and the supernatant was removed. The residual was re-extracted with 200 µL of water-saturated EA and centrifuged similarly. The two supernatants were combined to make about 400 µL total volume, and 10 µL of which was injected into the UPLC. 20 µL of 2xNES (0.2M NaOH, 4% w/v SDS and 6 mM EDTA) added to the ~20 µL residual. The resulting suspension was slowly mixed with 280 µL EA and then centrifuged at 13.5 K rpm for 10 min. 10 µL of the supernatant was injected into the UPLC. In addition, 25 µL of un-extracted homogenate was mixed with 25 µL 2xNES, then with 350 µL EA, centrifuged at 13.5 K rpm for 10 min and the supernatant was analyzed by UPLC.

## UPLC

About 10 µL of sample extract was injected into a Waters Acquity UPLC system which included a binary solvent manager, sample manager, photodiode array detector (PDA), fluorescence detector (FLR), column heater and an Acquity UPLC BEH C18, 1.7 µM, 2.1 × 100 mm column. The PDA was set to measure hemin absorbance at 398 nm and the FLR to measure fluorescence of protoporphyrin IX (PPIX) at 404 nm excitation and 630 nm emission. Solvent A was 0.2% aqueous formic acid while Solvent B was 0.2% formic acid in methanol. The flow rate at 0.40 mL per min at 60 ˚C for the total run time of 7 min. The following successive gradient settings for run time in min versus A: 0.0, 80%; 2.5, 1%; 4.5, 1%; 5, 80%. The solvent composition gradient settings were all linear. For standards, solutions of known concentrations of authentic hemin, PPIX dissolved in NES were extracted and then analyzed by UPLC as the samples.

## Serum analyses

Serum iron, total iron binding capacity (TIBC) and transferrin saturation were measured using the Stanbio Iron and TIBC kit (VWR, cat. number 10152–550). Serum ferritin was quantified using the mouse ferritin ELISA kit (Abcam, cat. number ab157713). All kits were used according to the manufacturer's protocol.

## $^{59}$Fe-labeled erythrocytes

$^{59}$FeCl$_3$ purchased from Perkin Elmer Life Sciences (cat. number NEZ037001MC) was mixed with sodium citrate (1:50 molar ratio in a total volume of 1 ml) and incubated for 1 hr at room temperature to make $^{59}$Fe-citrate. To generate $^{59}$Fe-labeled red blood cells (RBCs), adult donor mice were first injected intraperitoneally once per day for three consecutive days with 50 mg/kg of phenylhydrazine (Sigma Aldrich, cat. number 114715) to induce anemia. Thirty min after the last phenylhydrazine dose, donor mice were injected intraperitoneally with 200 µl of radiolabeled $^{59}$Fe-citrate (0.03 µM,~12 million cpm). Following a 3 day rest, donor mice were anesthesized (10% Ketamine with 5% Xylazine) and whole blood was collected by retro-orbital bleeding using heparinized tubes. Whole blood was mixed with an equal volume of Alsever's solution (Sigma Aldrich, cat. number A3551). Mice were then sacrificed by cervical dislocation. $^{59}$Fe-RBCs were collected by centrifugation and washed with DPBS before counting. $^{59}$Fe-RBCs were opsonized with the mouse red blood cell antibody (Rockland, cat. number 210–4139); 20 µl of antibody was used for approximately 10$^9$ RBCs. The suspension was diluted to 10 ml with PBS, and incubated at 37˚C on a rotating table for 20 min. The opsonized cells were washed twice with DPBS and counted again. The opsonized $^{59}$Fe-RBCs were diluted with DPBS and injected intraperitoneally into iron-deficient WT and KO mice (250 µl, 860,000 cpm per mouse). $^{59}$Fe-RBCs -injected mice provided with food and water were sacrificed by $CO_2$ asphyxiation at 24 and 96 hr post-injection. Whole blood was collected retro-orbitally for counting prior to sacrifice. Counts for separate tissues and spleen homogenates were collected on a

Perkin Elmer (Packard) Wizard gamma counter (Efficiency ~21%) and whole carcass counts were collected using a Model 2200 scalar ratemeter (Efficiency ~0.3%) (LUDLUM measurements, Inc, Sweetwater, TX).

To count for $^{59}$Fe in different extracts in spleens, spleens from the 96 hr time-point were dounce homogenized in lysis buffer (1% Triton X-100, 1% proteinase K, Tris-HCl pH 8.0, NaCl, CaCl$_2$) and incubated overnight at 37°C. One-third of the homogenate was centrifuged at 11,000 g for 45 min to obtain the insoluble fraction. Ethyl acetate (EA) (one part glacial acetic acid, four parts ethyl acetate) was added to the supernatant (one part supernatant, four parts EA). The suspension was vortexed for 1 min and centrifuged at 800 g for 2 min to separate the organic and aqueous phase. This extraction was repeated once to the aqueous layer and EA extracts were pooled for counting.

## Spleen and bone marrow cells isolation

A mouse spleen was cut up into 1–3 mm pieces and placed in 5 ml of dissociation buffer (RPMI, 1X collagenase B, 1X DNase I) in tubes with a magnetic stir bar. Tubes were placed on a magnetic plate and spleens were dissociated at 37°C for 45 min. Homogenates were passed through 70 μm filters and cells were pelleted by centrifugation at 800 g for 10 min. Bone marrow cells were flushed from the femur and tibia of mice using a syringe and 18G needle with 10 ml of FACS buffer (DPBS with 2% FBS). Cell aggregates were dissociated by pipetting. Cell suspensions were centrifuged at 800 g for 5 min and supernatants were discarded. When red blood cells (RBCs) were not needed for analysis, cells were resuspended in 1 ml of RBC lysis buffer (150 mM NH$_4$Cl, 10 mM NaHCO$_3$, 1.3 mM EDTA) and left at room temperature for 3 min. The lysis step was quenched by adding 5 ml of RPMI and cells were collected by centrifugation at 800 g for 10 min and resuspended in appropriate buffers for downstream application. From splenic and bone marrow cell suspensions, anti-F4/80 microbeads were used in conjunction with LS columns (Miltenyi Biotech, cat. number 130-110-443 and cat. number 130-042-401) according to the manufacturer's protocol. To isolate monocytes from bone marrow suspension, the mouse monocyte isolation kit was used (Miltenyi Biotech, cat. number 130-100-629). Monocytes and macrophages were cultured as described previously (*White et al., 2013*).

## Preparation of opsonized RBCs/beads and EP

Whole blood was collected by retro-orbital bleeding using heparinized tubes from mice and mixed with an equal volume of Alsever's solution (Sigma Aldrich, cat. number A3551). RBCs were opsonized with the mouse red blood cell antibody (Rockland, cat. number 210–4139); 20 μl of antibody was used for approximately $10^9$ RBCs.

For in vitro EP experiments, RBCs were added at a ratio of 1:10 (macrophage:RBCs) into cell culture medium and applied to BMDMs. One h later, medium was aspirated and cells were washed with DPBS and RBC lysis buffer (150 mM NH$_4$Cl, 10 mM NaHCO$_3$, 1.3 mM EDTA) before fresh medium was replaced.

## LDH assays

BMDMs were seeded in 6-well plates at a density of $4 \times 10^6$ cells per well and treated with the indicated treatments. Cells were harvested at indicated time points. LDH assays were conducted using a kit purchased from Sigma (Cat. number TOX7) and assay was conducted according to the manufacturer's protocol.

## ROS assays

BMDMs were seeded in 24-well plates at a density of $2 \times 10^5$ cells per well, treated with opsonized RBCs (1:10) and intracellular ROS was determined with a fluorometric intracellular ROS kit (Sigma-Aldrich, MAK142). Stained cells were visualized and captured using Leica DMI6000B with Cy5 filter set. The ImageJ software was applied to analyze regions of interest (ROIs) over multiple cells (n > 50). The fluorescence intensity from all the ROIs was averaged, and the corresponding background average was subtracted to yield the signal intensity for each condition.

## GSH quantification

BMDMs were seeded in 6-well plates at a density of $2 \times 10^6$ cells per well and treated with opsonized RBCs (1:10). Cells were harvested at indicated time points. LDH assays were conducted using a kit purchased from Abcam (Cat. number ab138881) and assay was conducted according to the manufacturer's protocol. The amount of GSH and GSSG were normalized to the protein concentration of each sample measured by a BCA protein assay.

## Heme quantification and immunoblot of lysates

BMDMs were lysed in a lysis buffer (1% Trition X-100% and 2% SDS 62.5 mM NaCl, 1 mM EDTA, 20 mM Hepes pH 7.4, 2X protease inhibitor) and sonicated. Heme in the lysate was quantified by adding 5 µl lysate to 200 µl 2M oxalic acid and heated at 95℃ for 30 min. A duplicate set of samples were kept at room temperature. Both sets were then measured for fluorescence at excitation and emission wavelengths of 400 nm and 662 nm, respectively. Values for the room temperature set were subtracted from that of the heated set and all measurements were normalized to the protein concentration of the corresponding samples. Lysates were then subjected to SDS-PAGE and immunoblotting by HMOX1 antibody (Enzo, cat. number ADI-SPA-896, 1:1000 dilution) and SLC48A1 antibody as mentioned previously.

## Immunofluorescence of LAMP1

Bone marrow isolated F4/80[+] macrophages (BMMs) were seeded onto coverslips and fixed with 4% PFA pH 7.4 for 40 min on ice, then washed twice with DPBS. Quenching was done using 0.1 M ethanolamine for 5 min and room temperature, twice. Coverslips were washed twice with DPBS and incubated in a buffer containing DPBS, 3% BSA, 0.4% saponin for 20 min at room temperature. Coverslips were then incubated in a buffer containing DPBS, 1% BSA and 0.15% saponin for 10 min before incubation with the LAMP1 primary antibody (Developmental Studies Hybridoma Bank, cat. number 1D4B, 1:100 dilution) for 1 hr at room temperature. Cells were then washed three times with DPBS and incubated with the secondary antibody in the same buffer (Invitrogen, cat. number A-2121, 1:2000 dilution) for 1 hr at room temperature. Coverslips were then washed three times with DPBS and stained with DAPI (1:30000 dilution) for 1 min before mounting with pro-long antifade (Thermofisher, cat. number P36930). Images were taken using the DeltaVision Elite Deconvolution microscope.

## Flow cytometry

Prior to staining for flow cytometry, cells were resuspended in FACS buffer and counted to ensure that appropriate amounts of antibodies would be added. Cells were stained in 500 µl FACS buffer with the respective antibodies on ice for 30 min. After staining, cells were centrifuged at 800 g for 5 min and washed with FACS buffer before being resuspended in FACS buffer for analysis. For erythroid cell populations in the spleen and bone marrow, T cells, B cells, platelets, megakaryocytes and neutrophils were stained with fluorescein isothiocyanate (FITC)-conjugated-CD4 (eBioscience, cat. number 14-0041-86), CD8a (eBioscience, cat. number 14-0081-86), B220 (eBioscience, cat. number 14-0452-86), CD41 (eBioscience, cat. number 14-0411-85), Gr-1 (eBioscience, cat. number 11-5931-82) and CD11b (eBioscience, cat. number 14-0112-86) and dump gating was used to exclude these cell populations during analysis. In addition, cells were stained with antibodies for allophycocyanin (APC)-conjugated Ter-119 (eBioscience, cat. number 47-5921-82) and eFluor-450-conjugated CD44 (eBioscience, cat. number 48-0441-82). For non-erythroid cell analysis, splenic cells were treated with RBC lysis buffer and stained with the following antibodies: phycoerythrin (PE)-conjugated Treml4 (Biolegend, San Diego, CA, cat. number 143304), APC-conjugated F4/80 (eBioscience, cat. number 17-4801-80), and PE-Cyanine7 (PE-Cy7)-conjugated CD11b (eBioscience, cat. number 25-0112-81). For all antibodies, 1 µl was used to stain $1 \times 10^6$ cells. Samples were run in a FACS Canto II or FACS Aria system (BD) and analysis was performed using FlowJo software (Tree Star Inc, Ashland, OR).

## Gating of erythoid populations

The gating strategy for identifying different erythroid populations was as described elsewhere (*Chen et al., 2009*), using CD44 as a marker for different populations.

## Transmission electron microscopy

Cells were seeded/cultured onto pre-cleaned Aclar disks as monolayers at a density of $1 \times 10^6$ cells per well. For fixation, the growing medium was replaced with cold fixative solution (2.5% glutaraldehyde, 1% PFA; 0.1 M Cacodylate buffer, pH 7.2). The cells were incubated at 4°C overnight. The aclar disks were then gently rinsed twice with cacodylate buffer and post-fixed with 2% Osmium tetroxyde for 1 hr at room temperature and pre-stained for 1 hr with saturated filtered uranyl acetate at room temperature. Dehydration and infiltration followed by using ethanol -graded series for 5 min each (50%; 70%; 95%−2X; 100%−3X) and pure acetone (3 × 5 min). The aclar disks were infiltrated with 50% epon resin:acetone for 1 hr, then with 75% epon resin for overnight and 100% epon resin over 8 hr with three changes. Cells were embedded and polymerized at 60°C for 24 hr. Ultrathin (70 nm) sections were obtained with diamond knife (Diatome) and an ultratome Leica UC6 (Leica Microsystems, Vienna, Austria). Grids with sections were post stained with saturated uranyl acetate in dH2O for 20 min and with lead citrate for 10 min and imaged at 120 kV using JEOL-JEM 1400 Plus electron microscope (Tokyo, Japan).

## Scanning electron microscopy

Hemozoin crystals were adhered to an aluminum stub with a carbon adhesive. The stubs were then coated in a sputter coater with a layer of gold/palladium to decrease charging of sample. Samples were examined using a F.E.I. Quanta 600 with field emission gun at 20KV and a working distance of approximately 10 nm.

## X-ray powder diffraction analysis

The powder samples were mounted in a 0.4 mmϕ glass capillary. The X-ray powder diffraction data were measured using a large Debye-Scherrer camera with an imaging-plate as a detector installed at SPring-8 BL02B2 (*Nishibori et al., 2001*). The wavelength of incident X-ray was 0.800 Å. The exposure time was 17 min. The X-ray powder diffraction patterns were collected in 0.01° steps in 2θ. The data range of the present analysis was from 2.0° to 30.0° in 2θ, which corresponds to more than 1.55 Å in d-spacing range.

Peak positions and relative intensities of powder profile were similar to those of β-hematin (*Straasø et al., 2011*). The structure of β-hematin reported by Tine et al was used for the initial model of Rietveld refinement. The rigid-body Rietveld analysis was carried out as an initial stage of refinement using the program SP (*Nishibori et al., 2007*). Overall isotropic thermal factor was used in the analysis. The position and orientation of molecules were refined in the analysis. The reliability factors of final Rietveld refinement were $R_{wp}$ = 2.6% and $R_I$ = 5.7%, respectively. The refined lattice constants are $a$ = 12.244(4) Å, $b$ = 14.734(3) Å, $c$ = 8.072(2) Å, $\alpha$ = 90.45(2)°, $\beta$ = 96.80(3)°, and $\gamma$ = 97.56(2)° with space group $P$-1.

## X-ray fluorescence microscopy (XFM)

Sample preparation was performed as described previously (*Bhattacharjee et al., 2016*). Briefly, macrophages were cultured directly onto sterilized silicon nitride membranes (1.5 × 1.5 mm, SiN, Silson Ltd, Northhampton, England) that had been incubated with sterile 0.01% Poly-L-lysine solution (Sigma-Aldrich, St Louis, MO). For the XFM experiments, the cells on the SiN membranes were fixed with 4% paraformaldehyde, rinsed sequentially with PBS, isotonic 100 mM ammonium acetate, DI water and air-dried.

XFM data were collected on the Bionanoprobe (*Chen et al., 2015*), beamline 9-ID-B, at the Advanced Photon Source, Argonne National Laboratory, Argonne, IL. The incident X-ray energy was tuned to 10 keV using a Si-monochromator, the monochromatic beam was focused to 80 × 80 nm using a Fresnel zone plate. The sample was placed at 15° to the incident X-ray beam and the resulting X-ray fluorescence was collected at 90° using an energy dispersive 4-element detector (Vortex ME-4, SII Nanotechnology, Northridge, CA). Elemental maps were generated by extracting, background subtracting, and fitting the fluorescence counts for each element at each point using the program MAPS (*Vogt, 2003*). The fluorescent photon counts were translated into μg/cm$^2$ using calibrated X-ray standards (AXO products, Dresden, Germany).

## Statistical analyses

All data are shown as means ± SEM unless otherwise stated. Means of groups were compared by using Student's unpaired t test. A $p$ value of < 0.05 was considered statistically significant. Analyses were performed using PRISM seven software (GraphPad).

## Acknowledgements

We thank Hector Bergonia for help with the heme/UPLC measurements; Stacie Anderson and Martha Kirby for assistance with flow cytometry analyses; Edward Case for $^{59}$Fe calibration and counting; Hidetaka Kasai and Shogo Kawaguchi for x-ray powder diffraction data collection; Paul Sigala for providing *Plasmodium falciparum* hemozoin; Si Chen at the Advanced Photon Source for assistance with the XFM experiments; and Susumu Tonegawa and Tracey Rouault for the *HMOX1* mice. This work was supported by funding from the National Institutes of Health DK85035 and ES025661 (IH); T32 GM080201 (RP); the Utah Center for Iron and Heme Disorders was supported by funding from DK110858 (JP); and the Intramural Program of the National Human Genome Research Institute (LJG, DB). We acknowledge use of the Advanced Photon Source at Argonne National Laboratory, supported by the Department of Energy, Office of Science, Office of Basic Energy Sciences, under contract no. DE-AC02-06CH11357. The synchrotron experiments were performed at SPring-8 BL02B2 with the approval of the Japan Synchrotron Radiation Research Institute (JASRI) as a Partner User (Proposal No. 2017A0074). The funders had no role in study design, data collection and analysis, decision to publish, or preparation of the manuscript.

## Additional information

### Competing interests

Iqbal Hamza: IH is the President and Founder of Rakta Therapeutics Inc (College Park, MD), which is a company involved in the development of heme transporter-related diagnostics. He declares no other competing financial interests. The other authors declare that no competing interests exist.

### Funding

| Funder | Grant reference number | Author |
| --- | --- | --- |
| National Institute of Diabetes and Digestive and Kidney Diseases | DK085035 | Iqbal Hamza |
| National Institute of Environmental Health Sciences | ES025661 | Iqbal Hamza |
| National Institute of Diabetes and Digestive and Kidney Diseases | DK110858 | John D Phillips |
| National Human Genome Research Institute | Intramural | Lisa Garrett David Bodine |
| Department of Energy | DE-AC02-06CH11357 | Martina Ralle |
| Japan Synchrotron Radiation Research Institute | 2017A0074 | Eiji Nishibori Hiroshi Sugimoto |

The funders had no role in study design, data collection and interpretation, or the decision to submit the work for publication.

### Author contributions

Rini H Pek, Xiaojing Yuan, Conceptualization, Formal analysis, Validation, Investigation, Methodology; Nicole Rietzschel, Jianbing Zhang, Ana Ribeiro, William Simmons, Jaya Jagadeesh, Hiroshi Sugimoto, Investigation, Methodology; Laurie Jackson, Resources, Validation, Investigation, Methodology; Eiji Nishibori, Formal analysis, Methodology; Md Zahidul Alam, Methodology; Lisa Garrett, Resources, Formal analysis, Validation, Investigation, Methodology; Malay Haldar, Formal

analysis, Investigation, Methodology; Martina Ralle, Resources, Formal analysis, Funding acquisition, Validation, Investigation, Methodology; John D Phillips, Conceptualization, Resources, Formal analysis, Validation, Investigation, Methodology; David M Bodine, Conceptualization, Resources, Formal analysis, Funding acquisition, Validation, Investigation, Methodology, Writing—review and editing; Iqbal Hamza, Conceptualization, Resources, Software, Formal analysis, Supervision, Funding acquisition, Validation, Investigation, Visualization, Methodology, Writing—original draft, Writing—review and editing

## Author ORCIDs

Rini H Pek ⓘ https://orcid.org/0000-0002-2570-7534
Xiaojing Yuan ⓘ http://orcid.org/0000-0002-0969-5748
Hiroshi Sugimoto ⓘ http://orcid.org/0000-0002-3140-8362
Iqbal Hamza ⓘ https://orcid.org/0000-0003-0045-0610

## Ethics

Animal experimentation: All animal protocols were approved by the Institutional Animal Care and Use Committee at the University of Maryland, College Park (IACUC Animal Study Protocol R-NOV-18-61).

## Decision letter and Author response

Decision letter https://doi.org/10.7554/eLife.49503.sa1
Author response https://doi.org/10.7554/eLife.49503.sa2

## Additional files

### Supplementary files

• Supplementary file 1. Table 1: SLC48A1/HRG1 Mutant alleles produced by CRISPR/Cas9. Table 2: Serum iron panel for WT and KO animals fed standard or 2ppm iron diet. Table 3: qRT-PCR analyses for iron/heme metabolism genes.

• Supplementary file 2. Contains tables of statistical analyses not included in the Transparent Reporting File.

• Transparent reporting form

## Data availability

All data generated or analysed during this study are included in the manuscript and supporting files.

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
