## [Decision Letter]

Thank you for submitting your article "Hemozoin produced by mammals confers heme tolerance" for consideration by *eLife*. Your article has been reviewed by three peer reviewers, including David Ginsburg as the Reviewing Editor and Reviewer #1, and the evaluation has been overseen by Harry Dietz as the Senior Editor. The following individual involved in review of your submission has agreed to reveal their identity: Daniel E Goldberg (Reviewer #2).

The reviewers have discussed the reviews with one another and the Reviewing Editor has drafted this decision to help you prepare a revised submission.

Summary:

This manuscript reports a careful and comprehensive characterization of mice genetically deficient in the heme transporter HRG1, demonstrating sequestration of crystalline hemozoin and impaired ability to respond to dietary iron deficiency. Surprisingly, the systemic and erythropoietic effects of the export block on iron recycling and erythropoiesis are relatively mild.

Essential revisions:

1) Even accepting that only 50% of iron flux in mice comes from recycling, the recycling block from HRG-1 KO appears only partial as it takes 8 weeks, i.e. almost 1.5 erythrocyte lifespans to separate the severity of anemia of wt and ko mice in the absence of dietary iron (Figure 4). The argument in the Discussion that it takes that long to deplete stores is unconvincing, especially in mice during their growth phase. As the authors can verify based on their own measurements, without recycling there is only enough storage iron in the liver and the spleen for less than 1 week of erythropoiesis. So there must be an important and perhaps major alternative heme recycling pathway, independent of HRG-1. This deserves additional analysis and comment.

2) The surprising interaction between the HRG-1 KO and heterozygous KO of heme oxygenase-1 is only superficially explored. The addition of the HO-1 KO to the HRG-1 KO does not add much to the hemozoin story other than eliciting a more dramatic phenotype, and at this stage is pure phenomenology in search of explanation. In the current model, heme oxygenase is in the cytoplasm, downstream of HRG-1. Although heterozygous HO-1 KO by itself is essentially wild-type, it appears to have synthetic lethality on the HRG-1 KO background. This is contrary to the simple model and unexplained in the current manuscript. Figure 6E shows that HO-1 undergoes differential proteolytic modification in the absence of HRG-1, raising the possibility that the HRG-1 and HO-1 physically or otherwise interact, potentially decreasing the bioactivity of HO-1 below the 50% expected from the heterozygous state and interfering with other functions HO-1 might have. At the very least this observation deserves a more focused discussion.

3) The genetic background of the studied mice should be more precisely defined. Gene targeting was performed in C57BL/6J X29SV/J mice and later intercrosses to generate heterozygous and homozygous knockout mice for further study included variable numbers of backcrosses into C57BL/6J. Thus, the studied animals represent variable mixtures of the 129SV/J and C57BL/6J genetic backgrounds. Were comparisons between mice of different genotypes restricted to littermates, to control for contributions from strain-specific genetic differences? The approach used to address this issue should be more completely described.

4) Some of the images were difficult to interpret, even on a high resolution screen. Improving image quality, selecting different images or quantifying images may help resolve this issue. For example, the HRG1 immunohistochemistry (Figure 1D) is confusing, with ample staining evident in both the wild type and knockout sections. How is staining corresponding to HRG1 distinguished from heme pigmentation? Is HRP the best stain for a heme defect? If the dark staining is HRG1, then why is it seen in the knockout sections? If it represents heme pigmentation, then why is it seen in some of the wild-type tissues? Would a color-contrasting immunoperoxidase substrate or immunofluorescence be easier to interpret?

[Editors' note: further revisions were requested prior to acceptance, as described below.]

Thank you for resubmitting your work entitled "Hemozoin produced by mammals confers heme tolerance" for further consideration at *eLife*. Your revised article has been favorably evaluated by Harry Dietz as the Senior Editor, and three reviewers, one of whom is a member of our Board of Reviewing Editors.

The manuscript has been improved but there are some remaining issues that need to be addressed before acceptance, as outlined below:

Point 4 was felt to be adequately addressed by the revised figures and point 3 (mouse genetic background) now adequately described in the revised Materials and methods. However, the reviewers did suggest that the authors specify the origin of their C57BL/6. At some places they are referred to as C57BL/6J and others as just C57BL/6. There are significant genetic differences between C57BL/6J, C57BL/6N, C57BL/6C, etc. and they are not interchangeable.

Major point 2 is addressed by a single more focused sentence in the Discussion however the reviewers would prefer to see the current HMOX1 data retained.

---

## [Author Response]

Essential revisions:1) Even accepting that only 50% of iron flux in mice comes from recycling, the recycling block from HRG-1 KO appears only partial as it takes 8 weeks, i.e. almost 1.5 erythrocyte lifespans to separate the severity of anemia of wt and ko mice in the absence of dietary iron (Figure 4). The argument in the Discussion that it takes that long to deplete stores is unconvincing, especially in mice during their growth phase. As the authors can verify based on their own measurements, without recycling there is only enough storage iron in the liver and the spleen for less than 1 week of erythropoiesis. So there must be an important and perhaps major alternative heme recycling pathway, independent of HRG-1. This deserves additional analysis and comment.

Our data indicate that dietary iron, and therefore tissue iron stores, is the key modifier of the anemia phenotypes associated with the SLC48A1 KO mouse. The KO mice were developed on a mixed genetic background (75% C57BL/6, 25% 129/SvJ). Published studies comparing the iron status of C57BL/6 and 129/SvJ mice on an iron-balanced diet have shown that 129/SvJ mice have more hepatic iron stores (Dupic et al., 2002; Wang et al., 2007). We recently generated KO mice on a pure C57BL/6 background and placed these mice on an iron-deficient diet post-weaning (P21). KO C57BL/6 mice showed severe anemia within 5.5 weeks compared to WT littermate controls, which approximates one lifespan of circulating erythrocytes (Author response image 1). This result indicates that genetic strain differences that influence body iron stores are important modifiers of the SLC48A1 mutant phenotype.

We agree that there might be an alternate, specific or non-specific/low affinity, heme recycling pathway independent of SLC48A1. However our results clearly indicate that this alternate pathway is unable to circumvent the loss of SLC48A1 during dietary iron deficiency. We are conducting RNAseq and proteomics-based experiments to identify this alternate pathway using mice and *C. elegans*.

2) The surprising interaction between the HRG-1 KO and heterozygous KO of heme oxygenase-1 is only superficially explored. The addition of the HO-1 KO to the HRG-1 KO does not add much to the hemozoin story other than eliciting a more dramatic phenotype, and at this stage is pure phenomenology in search of explanation. In the current model, heme oxygenase is in the cytoplasm, downstream of HRG-1. Although heterozygous HO-1 KO by itself is essentially wild-type, it appears to have synthetic lethality on the HRG-1 KO background. This is contrary to the simple model and unexplained in the current manuscript. Figure 6E shows that HO-1 undergoes differential proteolytic modification in the absence of HRG-1, raising the possibility that the HRG-1 and HO-1 physically or otherwise interact, potentially decreasing the bioactivity of HO-1 below the 50% expected from the heterozygous state and interfering with other functions HO-1 might have. At the very least this observation deserves a more focused discussion.

Based on the reviewer’s statement “The addition of the HO-1 KO to the HRG-1 KO does not add much to the hemozoin story […]”, we will be happy to remove the HMOX1 data from the manuscript, if the editor agrees.

Indeed, as the reviewer noted, we weren’t expecting to observe any additional phenotypes in the HMOX1 HET; SLC48A1 KO mice and were surprised to find the 40% embryonic lethality. The reviewer is astute in noticing the differential banding patterns of HMOX1 protein in the absence of SLC48A1 (Figure 6E). This has led us to hypothesize that SLC48A1 forms a ternary complex with HMOX1 (and other proteins) during erythrophagocytosis for efficient heme transfer, degradation, and iron release. While this explanation is speculative, a more focused sentence is added to the Discussion to explain the HMOX1 HET; SLC48A1 KO mice phenotype.

3) The genetic background of the studied mice should be more precisely defined. Gene targeting was performed in C57BL/6J X29SV/J mice and later intercrosses to generate heterozygous and homozygous knockout mice for further study included variable numbers of backcrosses into C57BL/6J. Thus, the studied animals represent variable mixtures of the 129SV/J and C57BL/6J genetic backgrounds. Were comparisons between mice of different genotypes restricted to littermates, to control for contributions from strain-specific genetic differences? The approach used to address this issue should be more completely described.

We apologize for the lack of clarity in the manuscript. All mice used for experiments in our manuscript were N_1_F_2_ generation derived from intercrosses between *HRG1^+/-^* heterozygotes. The relative proportion of genetic background was determined to be 75% C57BL/6 and 25% 129/SvJ by speed congenics using 5307 SNP Illumina Chip. Furthermore, WT mice are wildtype littermate controls of KO mice. Taken together, experimental animals used were obtained from the same generation to control for differences in the genetic strain. We have edited the experimental procedures section to include this information.

4) Some of the images were difficult to interpret, even on a high resolution screen. Improving image quality, selecting different images or quantifying images may help resolve this issue. For example, the HRG1 immunohistochemistry (Figure 1D) is confusing, with ample staining evident in both the wild type and knockout sections. How is staining corresponding to HRG1 distinguished from heme pigmentation? Is HRP the best stain for a heme defect? If the dark staining is HRG1, then why is it seen in the knockout sections? If it represents heme pigmentation, then why is it seen in some of the wild-type tissues? Would a color-contrasting immunoperoxidase substrate or immunofluorescence be easier to interpret?

While HRP substrates can react with tissue heme to provide false signal, our experimental methods overcame this problem by pre-treating the sections with hydrogen peroxide to eliminate background signals. To show that there is virtually no background staining, we have now added secondary antibody only staining controls (Figure 1—figure supplement 1C) for both WT and KO tissues. In addition, we performed SLC48A1 immunohistochemistry with secondary antibody conjugated to alkaline phosphatase, which stains red. SLC48A1 staining is clearly observed in WT tissues but is absent in KO tissue (Figure 1—figure supplement 1D).

[Editors' note: further revisions were requested prior to acceptance, as described below.]The manuscript has been improved but there are some remaining issues that need to be addressed before acceptance, as outlined below:Point 4 was felt to be adequately addressed by the revised figures and point 3 (mouse genetic background) now adequately described in the revised Materials and methods. However, the reviewers did suggest that the authors specify the origin of their C57BL/6. At some places they are referred to as C57BL/6J and others as just C57BL/6. There are significant genetic differences between C57BL/6J, C57BL/6N, C57BL/6C, etc. and they are not interchangeable.

The mouse strain is C57BL/6J and we have now edited the manuscript accordingly.

Major point 2 is addressed by a single more focused sentence in the Discussion however the reviewers would prefer to see the current HMOX1 data retained.

As per reviewers’ wishes, we agree to preserve the HMOX1 data in the manuscript (Figure 6).